# Isoform-specific localization of DNMT3A regulates DNA methylation fidelity at bivalent CpG islands

Massimiliano Manzo[1,2], Joël Wirz[1], Christina Ambrosi[1,2], Rodrigo Villaseñor[1], Bernd Roschitzki[3] & Tuncay Baubec[1,*]

## Abstract

DNA methylation is a prevalent epigenetic modification involved in transcriptional regulation and essential for mammalian development. While the genome-wide distribution of this mark has been studied to great detail, the mechanisms responsible for its correct deposition, as well as the cause for its aberrant localization in cancers, have not been fully elucidated. Here, we have compared the activity of individual DNMT3A isoforms in mouse embryonic stem and neuronal progenitor cells and report that these isoforms differ in their genomic binding and DNA methylation activity at regulatory sites. We identify that the longer isoform DNMT3A1 preferentially localizes to the methylated shores of bivalent CpG island promoters in a tissue-specific manner. The isoform-specific targeting of DNMT3A1 coincides with elevated hydroxymethylcytosine (5-hmC) deposition, suggesting an involvement of this isoform in mediating turnover of DNA methylation at these sites. Through genetic deletion and rescue experiments, we demonstrate that this isoform-specific recruitment plays a role in *de novo* DNA methylation at CpG island shores, with potential implications on H3K27me3-mediated regulation of developmental genes.

**Keywords** CpG islands; DNA methylation; DNMT3A; H3K27me3; Polycomb
**Subject Categories** Chromatin, Epigenetics, Genomics & Functional Genomics; Transcription
The EMBO Journal (2017) 36: 3421–3434

See also: **RR Meehan & S Pennings** (December 2017)

## Introduction

DNA methylation is a well-established epigenetic mark involved in gene regulation and genome stability. The importance of DNA methylation for mammalian genome function is apparent by the lethal phenotypes observed upon individual and combined knock-out of DNA methyltransferases, whereas DNMT1 and DNMT3B knock-outs are embryonic lethal and DNMT3A knock-outs are lethal 4 weeks after birth (Lei *et al*, 1996; Okano *et al*, 1999; Chen *et al*, 2003). Furthermore, aberrant deposition of DNA methylation is frequently observed in cancers (Baylin, 2005; Yan *et al*, 2011). Recent genome-wide initiatives explored the distribution of methylated CpGs in various cell types and tissues at single-base pair resolution (Lister *et al*, 2009; Stadler *et al*, 2011; Hon *et al*, 2013). These datasets were crucial in identifying the frequency and localization of methylated cytosines, and also to monitor changes in methylation during cellular transitions, including differentiation in healthy individuals (Schultz *et al*, 2015) or cellular transformation (Akalin *et al*, 2012; Hovestadt *et al*, 2014). Furthermore, the discovery of 5-hydroxymethylcytosine (5-hmC) as an additional modification of mammalian genomes and the characterization of TET-mediated DNA methylation removal (Kriaucionis & Heintz, 2009; Tahiliani *et al*, 2009; Kohli & Zhang, 2013) provide compelling evidence that DNA methylation is highly dynamic and undergoes constant turnover at regulatory sites (Stroud *et al*, 2011; Feldmann *et al*, 2013; Kohli & Zhang, 2013).

The *de novo* enzymes DNMT3A and DNMT3B are responsible for establishing DNA methylation, while DNMT1 is the maintenance methyltransferase responsible for propagation of this mark at CpG dinucleotides after DNA replication. In addition, the *de novo* DNMTs are responsible for the frequently occurring non-CpG methylation in mammalian genomes and contribute to maintenance of CpG methylation through filling up gaps after DNMT1 or counteracting active demethylation (Ramsahoye *et al*, 2000; Liang *et al*, 2002; Jackson *et al*, 2004; Arand *et al*, 2012). Recently, DNMT3C, a novel rodent-specific member of the *de novo* DNMT family, has been identified to regulate DNA methylation in the male germline (Barau *et al*, 2016). How DNMTs are correctly recruited to the genome in order to establish and maintain DNA methylation is not completely understood. In addition, splicing and alternative promoter usage gives rise to various catalytically active and inactive DNMT isoforms with tissue- and cancer-specific expression preferences (Chen *et al*, 2002; La Salle & Trasler, 2006; Gopalakrishnan *et al*, 2009; Duymich *et al*, 2016), revealing a complex regulation of DNA methylation through isoform variation. Previous studies have measured subcellular localization, catalytic activity, and targeting specificities of these isoforms in

1 Department of Molecular Mechanisms of Disease, University of Zurich, Zurich, Switzerland
2 Molecular Life Sciences PhD Program of the Life Sciences Zurich Graduate School, University of Zurich, Zurich, Switzerland
3 Functional Genomics Center Zurich, ETH and University of Zurich, Zurich, Switzerland
*Corresponding author. Tel: +41 44 635 5438; E-mail: tuncay.baubec@uzh.ch

various cell types and *in vitro* (Chen *et al*, 2002, 2003; Choi *et al*, 2011; Gordon *et al*, 2013; Duymich *et al*, 2016). However, the *in vivo* preferences of individual isoforms and their contribution to the DNA methylation landscape remains to be uncovered.

Based on numerous biochemical, structural, and genome-wide studies, the context-dependent crosstalk of DNA methylation with histone modifications, DNA sequence properties, transcriptional activity, or transcription factors became more and more apparent in the recent years (Rose & Klose, 2014; Ambrosi *et al*, 2017). Most importantly, histone modifications have been identified to play a major role in guiding DNA methylation (Rose & Klose, 2014; Du *et al*, 2015). For example, H3K4me3 prevents binding of the *de novo* methyltransferases to unmethylated CpG islands (Otani *et al*, 2009) and H3K36me3 was identified to enhance DNMT3B localization to actively transcribed gene bodies *in vivo* (Baubec *et al*, 2015). At the same time, DNA methylation has been shown to influence the deposition of histone modifications. For example, removal of DNA methylation results in spreading of the H3K27me3 mark from bivalent promoters (Lynch *et al*, 2011; Brinkman *et al*, 2012; Marks *et al*, 2012; Reddington *et al*, 2013; Jermann *et al*, 2014), and switching between H3K27me3- and DNA methylation-mediated repression has been described during development and in cancer (Schlesinger *et al*, 2006; Widschwendter *et al*, 2006; Mohn *et al*, 2008). This antagonism between DNA methylation and H3K27me3 suggests a dynamic crosstalk between these epigenetic marks that could be required for ensuring correct gene expression programs.

In this study, we explored the role of DNMT3A in establishing DNA methylation patterns in mouse ES and neuronal progenitor cells. More specifically, we focused our attention on the genomic location and DNA methylation activity of the longer DNMT3A isoform, DNMT3A1 (Fig 1A). We show that DNMT3A1 preferentially localizes to CpG islands bivalently marked by H3K4me3 and H3K27me3 in mouse embryonic stem cells, coinciding with the promoters of many transcription factors. This preference is further observed during neuronal differentiation, where DNMT3A1 binding follows H3K27me3 dynamics. DNMT3A1 binding to these sites coincides with elevated 5-hmC deposition at CpG island shores, and loss of DNMT3A results in reduced 5-hmC and erosion of DNA methylation at these sites. Re-expression of DNMT3A1, but not DNMT3A2, in *Dnmt-triple*-KO ES cells preferentially re-targets DNA methylation to bivalent CpG island shores and results in increased production of 5-hmC. These results demonstrate that the DNMT3A1 isoform is responsible for sustaining methylcytosine turnover at bivalent CpG islands.

# Results

## Isoform-specific localization of DNMT3A1 and DNMT3A2 to the genome of mouse ES cells

The *Dnmt3a* gene is transcribed from two alternative promoters, resulting in two protein isoforms: DNMT3A1 and DNMT3A2, respectively (Chen *et al*, 2002) (Fig 1A and Appendix Fig S1A). We have analyzed the differential expression of these two isoforms in detail during *in vitro* differentiation of murine ES cells (Appendix Fig S1B and C) and by using publicly available CAGE-seq data obtained from

various stages and tissues during mouse development (FANTOM5 Consortium, 2014). The CAGE-seq analysis indicates that the shorter isoform DNMT3A2 is mainly expressed during early development, whereas the longer DNMT3A1 isoform is expressed in the majority of samples (Fig 1B, Appendix Fig S1D and E, and Dataset EV1). Given these differences in isoform expression, we wanted to investigate the binding behavior of the DNMT3A isoforms and their potential contribution to DNA methylation. Standard chromatin immunoprecipitation (ChIP) for DNMT proteins is limited by the availability of suitable antibodies and furthermore does not allow to distinguish genomic binding preferences between protein isoforms. To overcome these limitations, we have utilized RAMBiO (Baubec *et al*, 2013), a biotin-tagging approach which we have previously employed to investigate genomic binding of DNMTs in mouse ES cells (Baubec *et al*, 2015). This approach allowed us to tag individual DNMT3A isoforms, express them from the same heterologous genomic site at similar levels, and perform stringent detection of DNMT3A binding by streptavidin-based ChIP followed by high-throughput sequencing (Appendix Fig S2A–E and Dataset EV2).

Using this approach, we have obtained binding maps for DNMT3A1 in mouse ES cells from two individual clones, and compared its binding to DNMT3A2 and DNMT3B1 (Fig 1C–F and Appendix Fig S3A and B). Similar to DNMT3A2 and DNMT3B, binding of DNMT3A1 is excluded from unmethylated and low methylated regions (LMR) that coincide with active proximal and distal regulatory elements (UMR and LMR, respectively, from Stadler *et al*, 2011), while fully methylated regions (FMR) are bound by DNMT3A1 in a methylation density-dependent manner (Fig 1C and Appendix Fig S3C and D). However, we observed several differences in DNMT3A1 binding when compared to the other DNMT3 proteins, suggesting that this isoform could display unique targeting preferences to the genome. This is evident from lack of binding to actively transcribed gene bodies occupied by DNMT3B (Baubec *et al*, 2015) (Fig 1D), differential localization at numerous sites in the genome (Fig 1E and Appendix Fig S3E), and from its genome-wide correlations to various chromatin features. The latter indicates preferential localization of DNMT3A1 to genomic sites enriched for H3K27me3 (Fig 1F and Appendix Fig S3F)—a feature not observed for DNMT3A2 or DNMT3B. This differential binding behavior of DNMT3A1 motivated us to explore the localization of this isoform in more detail.

## The DNMT3A1 isoform localizes to H3K27me3-positive CpG island shores

Based on the observed differences in genomic binding between the DNMT3A isoforms and DNMT3B, we aimed to identify individual and overlapping binding sites of these enzymes. Therefore, we have calculated statistically significant enriched regions for each individual DNMT protein (see Materials and Methods), resulting in 3,970 regions exclusively enriched for DNMT3A1, 3,838 for DNMT3A2, 3,432 for DNMT3B, and sites that are shared between the *de novo* DNMT proteins (Fig EV1A and Appendix Fig S4A). In order to extract the binding preferences of DNMT3A1, we have focused our attention on the exclusively bound regions only and characterized these by calculating enrichments for various chromatin modifications, transcription, and local sequence properties. We again

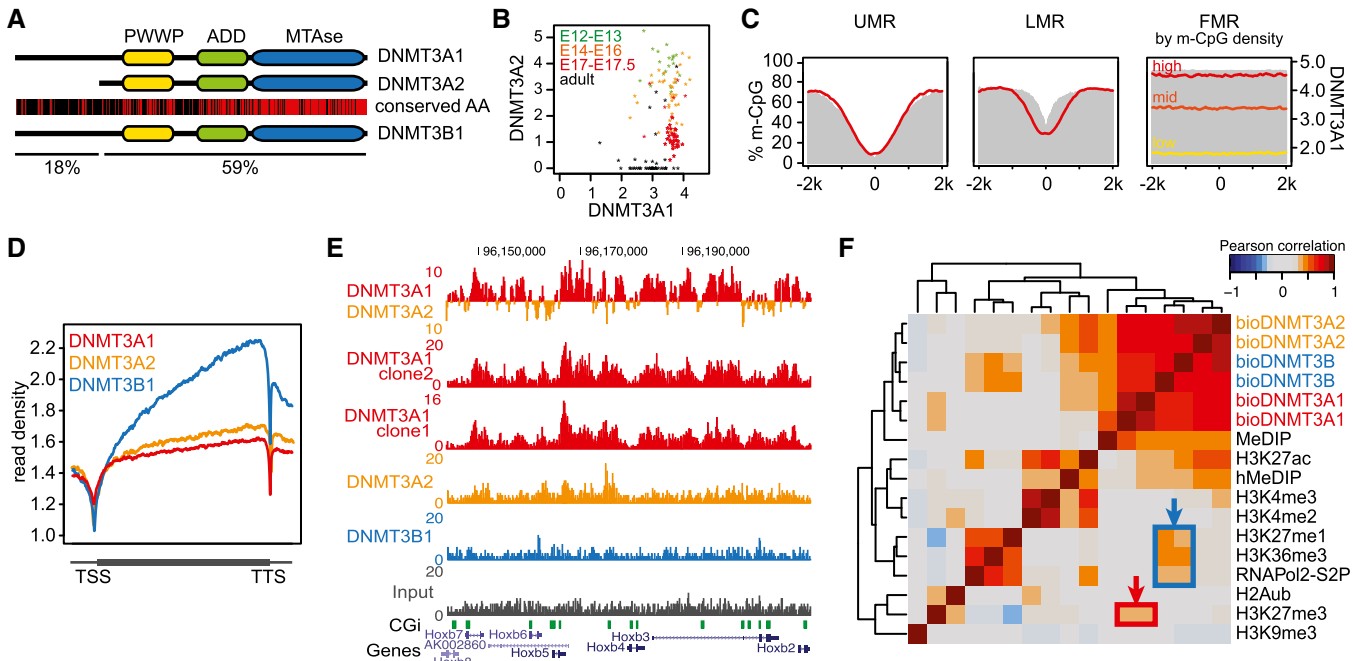

**Figure 1.  The DNMT3A isoforms DNMT3A1 and DNMT3A2 display differential localization along the mouse ES cell genome.**

A   Similarities in domain composition and amino acid sequence between murine DNMT3A and DNMT3B. Conserved amino acids are indicated as red bars. The longer and shorter DNMT3A isoforms DNMT3A1 and DNMT3A2 are shown.

B   Differential expression of DNMT3A isoforms measured by CAGE-seq data obtained from the FANTOM5 consortium. Indicated are data points obtained from various tissues during early mouse development at the indicated embryonic days (E) and from adult animals. Shown are $\log_2$-transformed CAGE-seq read counts overlapping the isoform-specific promoters (see also Appendix Fig S1D and E).

C   Average profiles indicating DNMT3A1 binding around genomic regions containing unmethylated, low methylated, and fully methylated regions (UMR, LMR, and FMR, respectively, from Stadler *et al*, 2011). DNA methylation at these sites is indicated in gray (% m-CpG); binding of DNMT3A1 is indicated in red (average read counts from ChIP-seq data). At FMRs, DNMT3A1 binding is shown at three FMR groups stratified by methyl-CpG-density to indicate binding dependency on the density of methylated CpGs (see also Appendix Fig S3C and D).

D   Comparative binding analysis between DNMT3A1, DNMT3A2, and DNMT3B indicates lack of DNMT3A recruitment to transcribed gene bodies. Shown are average density plots for ChIP-seq read counts around non-overlapping genes scaled by gene length.

E   Representative genome browser view exemplifying differences in binding between the *de novo* methyltransferases. Shown are read counts per 100 bp for ChIP-seq and input samples. Gene models, CpG islands, and repetitive elements from the UCSC genome browser are indicated below. Top track indicates differential binding between DNMT3A1 (red) and DNMT3A2 (orange).

F   Genome-wide cross-correlation analysis between *de novo* DNMT binding and various chromatin modifications. Pearson's correlations are calculated on $\log_2$-transformed read counts over 1-kb-sized tiles covering the mouse genome. Increased correlations between DNMT3A1 and H3K27me3, as well as for DNMT3B with transcriptionally active features, are indicated by red and blue boxes, respectively.

observed that DNMT3A1-bound sites are frequently enriched for H3K27me3, while DNMT3A2 sites do not display any preference, and DNMT3B sites are associated with transcribed regions marked by H3K36me3 as previously described (Baubec *et al*, 2015) (Fig 2A and B, and Appendix Fig S4A). Direct comparison between the antagonistic modifications H3K27me3 and H3K36me3 on a genome-wide scale further revealed the isoform-specific preference of DNMT3A1 for H3K27me3-bound sites (Fig 2C). Surprisingly, although DNA methylation levels measured directly at DNMT-binding sites are comparably high for all three proteins (> 80% average, Fig 2D and Appendix Fig S4B), regions immediately upstream and downstream of the DNMT3A1-binding sites display a sharp reduction in DNA methylation (Fig 2D, top). CpG density analysis revealed the opposite behavior, with increasing CpG density immediately upstream and downstream of DNMT3A1-binding sites (Fig 2D, bottom). This was not evident for DNMT3A2- or DNMT3B-bound sites, suggesting that genomic regions with

reduced DNA methylation and increased CpG density frequently occur near DNMT3A1-binding sites (Fig EV1B).

These results indicate a potential preference of the longer isoform DNMT3A1 for hypomethylated and CpG-rich regulatory regions decorated by H3K27me3. Therefore, we have focused our attention on the CpG island promoters that are bivalent for H3K4me3 and H3K27me3 marks (Mikkelsen *et al*, 2007; Mohn *et al*, 2008). By binning CpG island promoters into bivalent and non-bivalent promoters based on the presence of H3K27me3 (Appendix Fig S4C), we could observe that DNMT3A1 is mainly enriched at bivalent CpG islands (Figs 2E and EV1C). However, DNMT3A1 does not completely overlap with the H3K27me3 signal and is excluded from the center of the CpG island—in line with H3K4me3 preventing binding of *de novo* DNMTs through their ADD domains (Otani *et al*, 2009). Instead, DNMT3A1 localizes upstream and downstream of the CpG islands (Fig 2E and Appendix Fig S4D), coinciding with CpG island shores that have been reported to be differentially

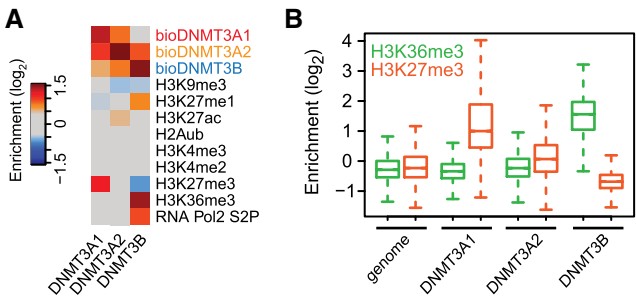

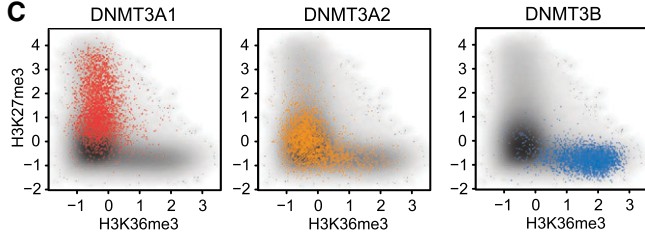

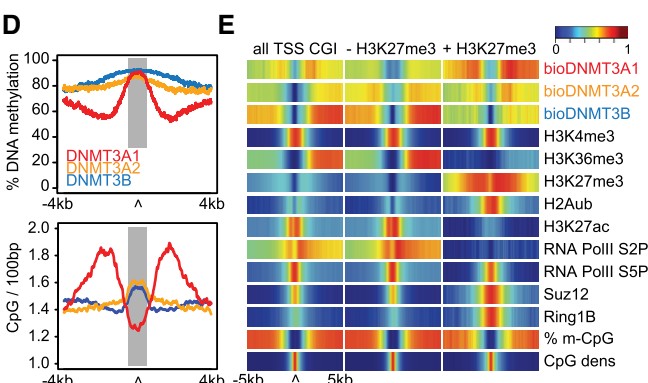

**Figure 2. The DNMT3A1 isoform preferentially localizes to H3K27me3-bivalent CpG islands.**

A   Enrichment analysis of chromatin features under DNMT3-protein-binding sites reveals increased H3K27me3 localization at sites bound by DNMT3A1. Shown are average log$_2$-enrichments of the indicated chromatin modifications and factors over input at sites exclusively bound by individual *de novo* DNMTs.

B   Box plots indicating log$_2$-enrichment over input for H3K27me3 and H3K36me3 at DNMT3A1-, DNMT3A2-, and DNMT3B-binding sites, and compared to average enrichment in the entire genome. Boxes denote the inter-quartile range (IQR) and whiskers 1.5 × IQR.

C   Scatter plots indicating the specific binding preference of DNMT3A1 for H3K27me3 and DNMT3B for H3K36me3. Shown are genome-wide enrichments over input for these histone modifications highlighting their mutual exclusiveness (gray) and the location of DNMT3A1 (red)-, DNMT3A2 (orange)-, and DNMT3B (blue)-bound sites along these genomic distributions.

D   DNA methylation levels and CpG densities differ around *de novo* DNMT-binding sites. Top: Shown are average methylation values for CpGs centered around DNMT3-binding sites (gray area). Bottom: Shown are CpG densities calculated as number of CpG dinucleotides per 100 bp at DNMT3-binding sites. Average densities are calculated for exclusive binding sites of DNMT3A1 (red), DNMT3A2 (orange), and DNMT3B (blue).

E   DNMT3A1 preferentially localizes upstream and downstream of bivalent CpG island promoters. Shown are heat map density profiles for DNMT3 protein binding and chromatin features around all promoter-associated CpG islands and CpG promoters separated by H3K27me3. CpG island promoters are oriented according to their downstream gene.

methylated during development and in cancer (Doi *et al*, 2009; Irizarry *et al*, 2009; Hodges *et al*, 2011). This specific localization is not observed for DNMT3B which is fully absent from CpG islands shores, and only to a minor degree for DNMT3A2 (Fig 2E and Appendix Fig S4D), indicating an isoform-dependent recruitment of DNMT3A to these sites. We furthermore characterized the Gene Ontology annotation of the genes neighboring DNMT3A1-bound sites using GREAT (McLean *et al*, 2010) in order to identify whether there is a preference for a specific set of genes. This analysis revealed that DNMT3A1 sites coincide with genes involved in transcriptional regulation, including numerous DNA sequence-specific transcription factors (Fig EV1D). This enrichment was evident only for DNMT3A1 sites, further supporting the specific association of this isoform with Polycomb-regulated promoters of genes involved in developmental and differentiation processes.

## DNMT3A1 coincides with H3K27me3 in a cell type-specific manner

In order to test whether DNMT3A1 binding to Polycomb CpG island shores is cell type specific, we followed the genome-wide binding of both DNMT3A isoforms during differentiation of ES cells expressing the biotin-tagged isoforms to homogenous populations of neuronal progenitors (NPs) (Appendix Fig S2B and C). We first contrasted DNMT3A isoform binding dynamics to changes in DNA methylation to identify their potential contribution to the reported *de novo* methylation of proximal and distal regulatory regions during neuronal differentiation (Mohn *et al*, 2008; Stadler *et al*, 2011). Therefore, we have identified genomic regions that gain DNA methylation in neuronal progenitors based on available whole-genome bisulfite sequencing data obtained from the same cellular system (Stadler *et al*, 2011) (Appendix Fig S5A and B), and visualized cell type-specific DNMT3A protein binding to these regions. Both DNMT3A isoforms show increased recruitment to *de novo* methylated sites in neuronal progenitors (Fig 3A and Appendix Fig S5C), suggesting that both isoforms are involved in establishment of DNA methylation during early differentiation.

H3K27me3 is differentially localized to the genome in a cell type-specific manner (Mikkelsen *et al*, 2007; Mohn *et al*, 2008), and we observed that binding of DNMT3A isoforms differs at sites positive for H3K27me3 in neuronal progenitors. DNMT3A1, but not DNMT3A2, follows H3K27me3 dynamics and preferentially relocates to the vicinity of H3K27me3 during differentiation (Fig 3B and C, and Appendix Fig S5D–G). This isoform-specific recruitment was observed at sites gaining H3K27me3 and at genomic regions that have lost H3K27me3 in neuronal progenitors, including *Hox* gene clusters (Appendix Fig S5F). In addition, sites that display cell type-specific binding of DNMT3A1 in conjunction with H3K27me3 dynamics are also frequently accompanied by changes in DNA methylation (Fig 3C and Appendix Fig S5F).

## Differential recruitment of DNMT3A1 is mediated by its variant N-terminal region

We next wanted to explore how the genomic binding preference of DNMT3A1 is mediated toward H3K27me3 sites. First, we asked whether differential binding is caused by protein–protein interactions that are specific only to one of the two isoforms. We therefore

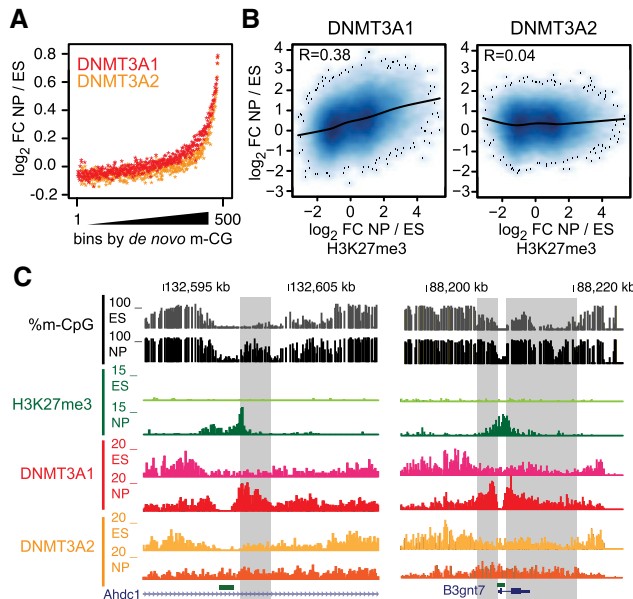

**Figure 3.  Dynamic colocalization of DNMT3A1 with H3K27me3 during neuronal differentiation.**

A  Both DNMT3A isoforms dynamically localize to *de novo* methylated regions during neuronal development. Increased DNMT3A binding in neuronal progenitors (NP) is shown as log₂-fold difference between ES and NP cells (y-axis) and is ranked based on *de novo* DNA methylation in NPs (x-axis). Individual data points denote median log₂-fold changes calculated from 500 consecutive windows ranked by DNA methylation.

B  Scatter plots indicate that genome-wide dynamics in DNMT3A1 binding correlates with regions that change H3K27me3 during neuronal differentiation. Shown are log₂-fold changes between ES and NP ChIP-seq signals of DNMT3A isoforms and H3K27me3 calculated at 1-kb windows positive for H3K27me3 in ES and/or NP cells.

C  Genome browser examples for regions that change DNMT3A1 binding and DNA methylation according to H3K27me3. Shown are read counts per 100 bp for ChIP-seq datasets and percentage of DNA methylation per individual CpGs. Regions that show concomitant changes in DNMT3A1 and DNA methylation are highlighted in gray.

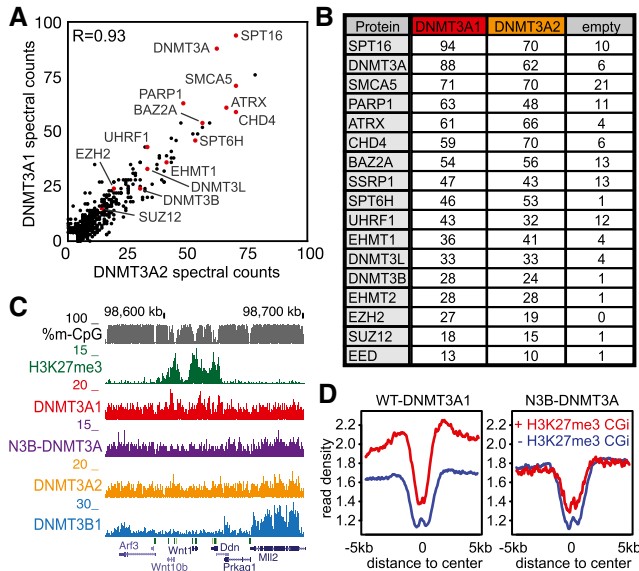

**Figure 4.  The N-terminal part of DNMT3A1 is required for localization to H3K27me3 sites.**

A  Scatter plot indicating similarities in protein–protein interactions between DNMT3A1 and DNMT3A2. Shown are the unique spectral counts per protein obtained from co-immunoprecipitation experiments performed in cell lines expressing biotin-tagged DNMT3A1 or DNMT3A2. The Pearson's correlation coefficient is shown. Red dots indicate proteins listed in (B).

B  Table indicating top-scoring proteins identified to interact with DNMT3A1 or DNMT3A2 in mouse ES cells. Shown are the results from (A) and an untagged cell line (empty) that serves as background control.

C  Genome browser example for regions indicating reduced binding of the chimeric N3B-DNMT3A protein to H3K27me3 sites. Shown are read counts per 100 bp for ChIP-seq datasets and percentage of DNA methylation per individual CpGs.

D  Average density plot around H3K27me3-positive and H3K27me3-negative CpG island promoters indicates reduced binding of the chimeric N3B-DNMT3A protein to H3K27me3 sites (see Appendix Fig S6B).

performed co-immunoprecipitation experiments from nuclear extracts in embryonic stem cells expressing the biotinylated DNMT3A isoforms as baits. Mass spectrometric detection of enriched proteins identified already-reported protein interactions of DNMT3A, including DNMT3L (Hata *et al*, 2002), DNMT3B (Li *et al*, 2007), EHMT1, and EHMT2 (Chang *et al*, 2011) (Fig 4A and B, and Dataset EV3). However, this analysis did not reveal any drastic differences in protein–protein interactions between the two DNMT3A isoforms. While the PRC2 components EZH2, SUZ12, and EED could be detected in the enriched fraction (Fig 4A and B), these were similar for both isoforms, suggesting that isoform-specific DNMT3A interactions with PRC2 members do not contribute to the observed genome-wide binding preference.

Based on the previous reports, the variant N-terminal regions of DNMT3A and DNMT3B are involved in mediating chromatin and DNA interactions (Jeong *et al*, 2009; Suetake *et al*, 2011; Baubec *et al*, 2015), we wanted to investigate the requirement of this N-terminal region for the observed localization of DNMT3A1 to H3K27me3 sites. We therefore have generated ES cell lines expressing a chimeric version of DNMT3A1 where we have replaced

the N-terminal part with that from DNMT3B (N3B-DNMT3A, Appendix Fig S6A). Using this cell line, we have repeated the genome-wide binding experiments in ES cells and measured the localization of the chimeric N3B-DNMT3A protein to CpG island promoters decorated by H3K27me3 and to genomic sites preferentially bound by the wild-type DNMT3 proteins. We observed that replacing the N-terminal part of DNMT3A1 reduces its preference for H3K27me3-positive CpG island promoters and for DNMT3A1-bound sites (Fig 4C and D, and Appendix Fig S6B and C). This indicates that the N-terminal part is required for the observed genome-wide binding preference of DNMT3A1 to Polycomb-regulated CpG islands. However, addition of the DNMT3B N-terminal end to DNMT3A did not result in increased localization of the chimeric N3B-DNMT3A protein to gene bodies (Appendix Fig S6D).

**DNMT3A1 regulates DNA methylation at bivalent CpG islands**

Previous studies have already highlighted that CpG island shores are differentially methylated in various cell types and cancers (Irizarry *et al*, 2009; Hodges *et al*, 2011). Therefore, the dynamic recruitment of DNMT3A1 to the vicinity of Polycomb-associated CpG islands

suggests a role for this isoform to demarcate the correct transition between methylated and unmethylated DNA at CpG island borders. We have utilized whole-genome bisulfite sequencing data from mouse ES cells (Stadler *et al*, 2011) to precisely map the transition from fully methylated (FMR) to unmethylated regions (UMR) around CpG island promoters (Appendix Fig S7A–C). This allowed us to better visualize DNMT3A1 localization to H3K27me3-positive UMRs and compare it to various other chromatin features at these sites (Appendix Fig S7D–F). This analysis revealed that 5-hmC is predominantly enriched at borders of H3K27me3-positive UMRs that are co-occupied by DNMT3A1 (Fig 5A and B, and Appendix Fig S7E). Clustering analysis of UMRs based on DNMT3A1 binding further reveals co-occurrence of DNMT3A1 with H3K27me3 and Polycomb group proteins at transcriptionally inactive genes, as well as site-specific colocalization with 5-hmC (Figs 5C and EV2A–D). TET1, the protein involved in oxidation of methylcytosine, is ubiquitously localized at all UMRs (Fig 5A and B).

The increased abundance of 5-hmC at Polycomb-regulated CpG islands coincides with preferential recruitment of DNMT3A1 to CpG island shores and suggests an involvement of DNMT3A1 in *de novo* DNA methylation at these sites. In order to directly test the requirement of DNMT3A, we have knocked out the *Dnmt3a* gene by using CRISPR/Cas9 (Appendix Fig S8A) and measured 5-hmC levels in these cell lines. This revealed a strong reduction in the 5-hmC signal at the borders of bivalent UMRs in the absence of DNMT3A (Fig 5D and Appendix Fig S8B and C), indicating that DNMT3A generates the substrate for TET-mediated oxidation at CpG island shores. To follow up on this potential role, we have measured DNA methylation at these sites using reduced-representation bisulfite sequencing (RRBS) in wild-type and *Dnmt3a* KO ES cells. Analysis of DNA methylation at the UMR flanks reveals that absence of DNMT3A leads to a reduction in DNA methylation at H3K27me3-positive UMRs (Fig 5E and Appendix Fig S8D and E).

However, these experiments were performed in a cell line where both DNMT3A isoforms have been deleted (Appendix Fig S8A).

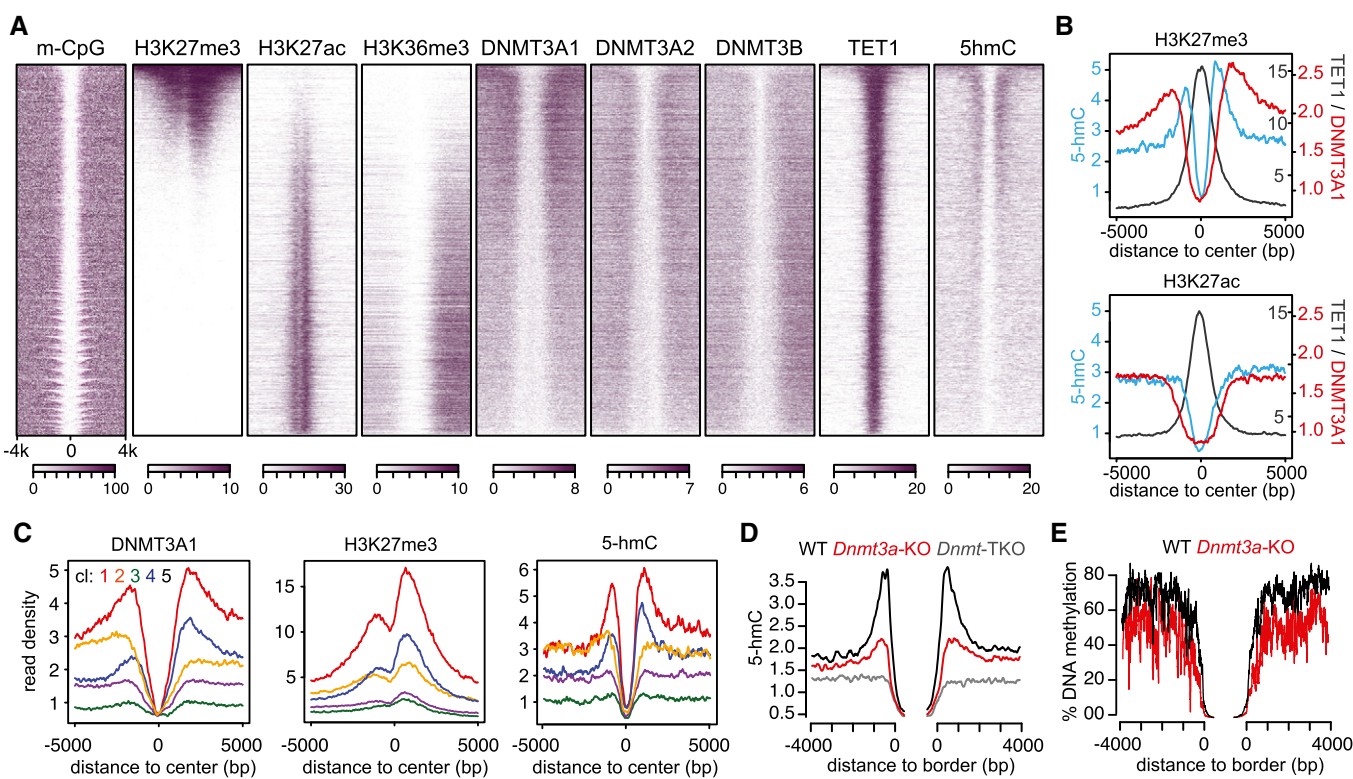

**Figure 5. Binding of DNMT3A1 to CpG island shores is required for DNA methylation regulation and promotes 5-hmC deposition.**

A   Heat map profiles of DNMT protein binding and chromatin features around promoter-associated unmethylated regions (UMR) oriented by gene direction and ranked by H3K27me3 show a preferential association of DNMT3A1 with H3K27me3-decorated UMRs. Furthermore, H3K27me3-positive UMRs are enriched for 5-hmC signals flanking the unmethylated center.

B   Average density profiles for top 25% of UMRs ranked by H3K27me3 indicate a localized 5-hmC signal at borders of UMRs that is localized between DNMT3A1- and TET1-binding sites (top). Top 25% of UMRs ranked by H3K27ac lack DNMT3A1 enrichment and have a more uniform 5-hmC signal (bottom). Shown are library-normalized read counts.

C   Average density profiles for UMR regions clustered based on DNMT3A1 binding. Five clusters were defined by k-means clustering that show distinct DNMT3A1 binding (d 1–5, see also Fig EV2 for more details). Clusters with highest DNMT3A1 binding are also enriched for H3K27me3 and also display elevated 5-hmC signals at UMR borders.

D   Average density plots centered over UMR borders identify a sharp signal in 5-hmC at the transition from FMR to UMR states in wild-type cells. This signal is reduced in ES cells lacking DNMT3A and is absent in the *Dnmt*-TKO cell line.

E   DNA methylation is reduced around bivalent CpG islands in DNMT3A-KO ES cells. Shown is the average DNA methylation measured by RRBS around UMR borders.

Consequently, it did not allow us to directly address isoform-specific DNA methylation, and furthermore, we cannot exclude that the remaining DNMT3B and DNMT1 proteins could partially compensate for the loss of DNMT3A at these sites. To directly test whether DNMT3A1 is indeed the main isoform responsible for setting DNA methylation at these sites, we have introduced DNMT3A1 into *Dnmt*-TKO cell lines that lack DNA methylation and 5-hmC, and performed whole-genome bisulfite sequencing (Appendix Fig S9A and B). In this situation, *de novo* methylation is only guided by DNMT3A1 binding preference, and in the absence of maintenance activity, DNA methylation should be only detected at sites preferentially targeted by DNMT3A1. Indeed, we observe that DNMT3A1-enriched regions are preferentially methylated compared to sites bound by DNMT3A2 or DNMT3B (Fig 6A), supporting that *de novo* methylation by DNMT3A1 is largely regulated by its targeting preference. Despite the increased methylation at sites directly bound by DNMT3A1, we again noticed that the methylation signal is reduced upstream and downstream of the DNMT3A1-binding sites due to the presence of neighboring CpG islands (Fig 6A), reminiscent of our observations obtained in wild-type cells (Fig 2D). Since DNMT3A1 is enriched at bivalent CpG island shores, we have directly analyzed the *de novo* DNA methylation rate at these sites. Density profiles surrounding H3K27me3-positive UMRs revealed an increased preference of *de novo* methylation in TKO cells expressing DNMT3A1 (Fig 6B and Appendix Fig S9C–E). This preference was not observed upon expression of the shorter DNMT3A2 variant in the *Dnmt*-TKO background (Fig 6B and Appendix Fig S9C–E), indicating that DNMT3A1 is the main isoform involved in regulating DNA methylation at bivalent CpG islands.

## DNMT3A1 recruitment promotes increased m-CpG turnover around H3K27me3 sites

To test whether the DNMT3A isoform-specific *de novo* methylation of cytosines is further oxidized by the TET enzymes, we have measured 5-hmC in the TKO cells expressing individual DNMT3A isoforms. These experiments revealed that re-expression of DNMT3A1, but not DNMT3A2, leads to an increased 5-hmC production around bivalent CpG islands (Fig 6C and D, and Appendix Fig S9F), demonstrating that the enzymatic activity of the DNMT3A isoform is providing the substrate for TET-mediated oxidation of methylated cytosines at these sites. 5-hmC is further processed by the TET enzymes to 5-formyl- and 5-carboxyl-methylcytosine (5-fC and 5-caC, respectively)—intermediates that are rapidly removed by the DNA repair enzyme TDG (Ito *et al*, 2011; He *et al*, 2011). Previous studies have indicated an accumulation of 5-fC and 5-caC at enhancers and H3K27me3-decorated promoters after knock-out or knock-down of TDG, revealing that DNA methylation turnover is actively occurring at these regulatory elements (Shen *et al*, 2013; Song *et al*, 2013). By re-analyzing these datasets, we observed an increase in 5-fC and 5-caC signals at DNMT3A1-enriched sites in the absence of TDG (Appendix Fig S10A and B), further supporting that binding of DNMT3A1 primarily occurs at sites with elevated turnover of cytosine methylation.

We have further investigated whether isoform-specific contribution of DNMT3A to DNA methylation turnover also occurs outside of Polycomb CpG island promoters. Toward this, we calculated the genome-wide differences in *de novo* DNA methylation activity and

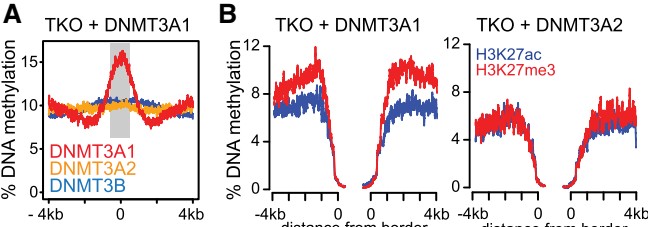

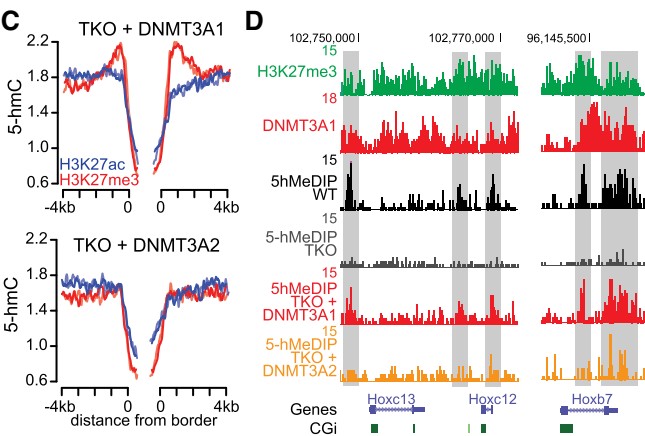

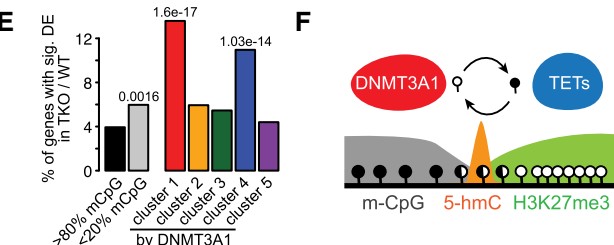

**Figure 6.  *De novo* DNA methylation activity of DNMT3A1 is directed to bivalent CpG island shores.**

A    Analysis of *de novo* DNA methylation in *Dnmt*-TKO cells re-expressing DNMT3A1 shows preferential DNA methylation activity at DNMT3A1-binding sites. Exclusive binding sites from DNMT3A1, DNMT3A2, and DNMT3B are shown in red, orange, and blue, respectively.

B    Expression of DNMT3A1, but not DNMT3A2, in *Dnmt*-TKO cells results in preferential *de novo* DNA methylation around bivalent CpG islands. Density plots show average methyl-CpG percent calculated from WGBS data around UMRs binned based on H3K27me3 or H3K27ac enrichment (top 25% ranked UMRs).

C, D    Re-expression of DNMT3A1 but not DNMT3A2 results in 5-hmC accumulation at bivalent CpG island shores. (C) Average density plots indicating 5-hmC read counts at UMR borders separated by H3K27me3 or H3K27ac from two independent replicates. (D) Representative genome browser views.

E    Percentage of genes with significant increase in gene expression in *Dnmt*-TKO over wild-type cells (FDR < 0.001 & log₂-FC > 0.5) according to their promoter classification. Promoters with more than 80% DNA methylation in wild-type cells (black); promoters with < 20% DNA methylation in wild-type cells (gray); or unmethylated CpG island promoters falling into one of five clusters based on DNMT3A1 binding (see Figs 5C and EV2A). *P*-values calculated based on hypergeometric tests are shown only for fractions with *P*-value < 0.05.

F    DNMT3A1 is preferentially recruited to the shores of bivalent CpG islands to maintain DNA methylation in the presence of TET-mediated oxidation of methyl-cytosines to 5-hmC.

5-hmC accumulation between TKO cells expressing DNMT3A1 or DNMT3A2, and compared these with chromatin features (Appendix Fig S10C). In general, we did not observe additional isoform-specific preferences toward other genomic features besides H3K27me3-decorated sites. In addition, enhancers that have been reported to display elevated DNA methylation turnover (Stroud *et al*, 2011; Feldmann *et al*, 2013) do not show any isoform-specific differences in *de novo* DNA methylation or 5-hmC accumulation (Appendix Fig S10D).

The site-specific recruitment of DNMT3A1 to Polycomb CpG island shores could have implications for H3K27me3 deposition at CpG island promoters. In the absence of DNA methylation, H3K27me3 has been reported to spread into neighboring regions, resulting in depletion of H3K27me3 from CpG island centers and influencing activity of Polycomb-regulated genes (Fouse *et al*, 2008; Lynch *et al*, 2011; Brinkman *et al*, 2012; Marks *et al*, 2012; Reddington *et al*, 2013; King *et al*, 2016). By reinvestigating published datasets generated in *Dnmt*-TKO and *Dnmt3a/3b*-DKO cells (King *et al*, 2016), we indeed observe that the CpG island promoters associated with enriched DNMT3A1 binding display the strongest reduction in H3K27me3 (Fig EV3A–C). And, as previously reported, by re-expressing DNMT3A1 in *Dnmt3a/3b*-DKO cells, H3K27me3 distribution at these promoters is more effectively restored than by DNMT3A2 or DNMT3B (King *et al*, 2016) (Fig EV3B and C). In addition to the H3K27me3 dynamics, we further detect increased DNaseI hypersensitivity and H3K27 acetylation in the absence of DNA methylation in *Dnmt*-TKO cells at the same DNMT3A1-bound and unmethylated CpG island promoters (Fig EV3D and E), coinciding with frequent deregulation of the associated genes (Fig 6E).

## Discussion

The *de novo* DNA methyltransferases DNMT3A and DNMT3B play an important role in mediating site-specific DNA methylation and are required to counteract active and passive removal of DNA methylation (Jackson *et al*, 2004; Goll & Bestor, 2005; Baubec *et al*, 2015; Ambrosi *et al*, 2017). However, the individual or overlapping roles of these enzymes in DNA methylation regulation are still not fully understood. In part, this is due to the presence of various isoforms and their differential expression, which inherently complicates their analysis. In this study, we have identified the preferential localization and activity of DNMT3A to H3K27me3-positive CpG islands in ES and neuronal progenitor cells. DNMT3A binding does not completely overlap with the H3K27me3 signal over the CpG island, but is restricted to its shores—the methylated regions outside of the island (Irizarry *et al*, 2009). This localization is in line with repulsion of the *de novo* methyltransferases by H3K4me3 and depletion of DNA methylation from the center of CpG islands (Otani *et al*, 2009), as well as with the reported role of the Polycomb-associated protein KDM2B in protecting bivalent islands from DNA methylation (Boulard *et al*, 2015). Importantly, this binding is mainly observed for the larger isoform DNMT3A1, whereas the shorter version, DNMT3A2, displays a more global binding and activity throughout the genome.

We show that DNMT3A1 dynamically localizes to H3K27me3 during cellular differentiation, suggesting a recruitment mechanism that is associated with H3K27me3 deposition. If and to what extent H3K27me3 is directly involved in the recruitment of DNMT3A1 remains to be understood. Previous studies have reported interactions between PRC2 complex members and DNMTs (Viré *et al*, 2005; Neri *et al*, 2013), and we observe similar interactions in our own study. However, deletions of key PRC1 or PRC2 components did not lead to changes in DNA methylation (Hagarman *et al*, 2013; Boulard *et al*, 2015) and we do not detect differential interactions of the DNMT3A isoforms with PRC2 component members, arguing against a role of PRC members in recruiting DNMT3A1 through protein–protein interactions. This suggests that the observed localization could also be facilitated by unknown chromatin or DNA features coinciding with Polycomb-regulated elements. Alternatively, repulsion of DNMT3A1 could be more efficient around non-bivalent CpG islands, either through increased presence of H3K4me3 or through transcriptional activity. By engineering a chimeric DNMT3A protein that contains the N-terminal part of DNMT3B, we show that the preferential localization of DNMT3A1 to H3K27me3 is reduced, suggesting a requirement of the DNMT3A1 N-terminal part in specifying this localization, in line with previous studies reporting DNA binding activity of this domain *in vitro* (Suetake *et al*, 2011). Nevertheless, this N-terminal domain replacement does not increase the affinity of DNMT3A to H3K36me3, and we do not observe any H3K36me3-dependent localization for both DNMT3A isoforms *in vivo*—despite the presence of a PWWP domain that binds H3K36me3 *in vitro* (Dhayalan *et al*, 2010). Further studies should bring insight into the mechanisms that lead to paralog- and isoform-specific genomic targeting preferences of the *de novo* DNA methyltransferases to H3K27me3 and H3K36me3 sites.

We further observe that DNMT3A1 localization to the shores of Polycomb-regulated CpG islands coincides with elevated 5-hmC deposition. Local enrichment of 5-hmC is indicative of an enhanced turnover of methylated cytosines by the activities of TET and DNMT enzymes at regulatory elements (Wu *et al*, 2011a; Pastor *et al*, 2011; Williams *et al*, 2011; Kohli & Zhang, 2013; Feldmann *et al*, 2013; Neri *et al*, 2015). Here, we report that the production of 5-hmC at H3K27me3-positive CpG islands is a consequence of DNMT3A1 recruitment to the CpG island shores. Loss of DNMT3A results in reduction in 5-hmC and subsequently erosion of methylated cytosines around CpG islands, establishing DNMT3A as a regulator of DNA methylation around these sites. This activity is in line with previous observations made in hematopoietic stem cells (HSC), where loss of DNMT3A resulted in expansion of large UMR, termed "canyons" (Jeong *et al*, 2014)—highlighting that the delicate balance between DNMT3A and TET activities is required to shape the methylome in multiple tissues and cell types. By expressing individual DNMT3A isoforms in *Dnmt*-TKO cells lacking DNA methylation and 5-hmC, we furthermore confirm that recruitment of the DNMT3A1 isoform is sufficient for preferential re-setting of DNA methylation at Polycomb CpG island shores. This enhanced *de novo* methylation activity around Polycomb CpG islands is, however, not observed for DNMT3A2, which acts as a global methylator—further suggesting a specialization of the DNMT3A1 isoform. Moreover, re-setting of DNA methylation by DNMT3A1 results in 5-hmC accumulation, substantiating that recruitment of DNMT3A1 is required for DNA methylation homeostasis at CpG island shores (Fig 6F).

It is well established that DNA methylation plays an important role in restricting H3K27me3 to unmethylated CpG islands. This is evident from preferential localization of H3K27me3 to endogenous

and ectopic unmethylated CpG-rich regions (Tanay *et al*, 2007; Mikkelsen *et al*, 2007; Mohn *et al*, 2008; Mendenhall *et al*, 2010; Jermann *et al*, 2014; Wachter *et al*, 2014), the identification of PRC complex members that preferentially bind unmethylated CpGs and GC-rich sequences (Farcas *et al*, 2012; Wu *et al*, 2013; Li *et al*, 2017), and numerous studies that reported spreading of H3K27me3 in the absence of DNA methylation (Lynch *et al*, 2011; Brinkman *et al*, 2012; Marks *et al*, 2012; Reddington *et al*, 2013; Jermann *et al*, 2014; King *et al*, 2016). We suggest that the balance between DNMT3A1 and TET activity is required to fine-tune the distribution of methylated and unmethylated CpGs and for restricting Polycomb domain boundaries to the promoters of developmentally regulated genes. This is supported by the increased re-setting of H3K27me3 in *Dnmt3a/3b*-DKO reconstituted with DNMT3A1 (King *et al*, 2016).

This supports the notion that DNA methylation can indirectly influence gene activity through instructing H3K27me3 deposition and that deregulation of this crosstalk could lead to disturbed execution of gene expression programs. Supporting evidence comes from increased deregulation of Polycomb-target genes in the absence of DNA methylation (Fouse *et al*, 2008; Wu *et al*, 2010; King *et al*, 2016), while in TET-knock-out ES cells, increased DNA methylation at Polycomb CpG islands results in derepression of the associated genes (Wu *et al*, 2011b; Lu *et al*, 2014; Kong *et al*, 2016). Furthermore, the crosstalk between DNA methylation and H3K27me3 is frequently disturbed in cancers. Several Polycomb-regulated promoters become hypermethylated during cancer progression, potentially resulting in a more stable, long-term repression by DNA methylation (Schlesinger *et al*, 2006; Widschwendter *et al*, 2006; Gal-Yam *et al*, 2008). And in leukemias, loss of DNMT3A activity is associated with hypomethylation around CpG islands and deregulation of *Hox* genes (Yan *et al*, 2011; Qu *et al*, 2014; Spencer *et al*, 2017). Our study identifies an isoform-specific function of DNMT3A at Polycomb sites that offer novel insights into the mechanisms underlying these epigenetic dynamics and provide potential targets for future therapies.

# Materials and Methods

### Cell culture and neuronal differentiation

Mouse embryonic stem cells were cultured on 0.2% gelatine-coated dishes in DMEM (Invitrogen) supplemented with 15% fetal calf serum (Invitrogen), 1× non-essential amino acids (Invitrogen), 1 mM L-glutamine, leukemia inhibitory factor, and 0.001% β-mercaptoethanol. Differentiation was performed as previously described (Bibel *et al*, 2007). $4 \times 10^6$ mES cells were cultivated in non-adherent bacterial dishes for 4 days for the formation of cellular aggregates in DMEM supplemented with 10% fetal calf serum (Invitrogen), 1× non-essential amino acids (Invitrogen), 1 mM L-glutamine, and 0.001% β-mercaptoethanol. Cellular aggregates were further cultivated in the same medium supplemented with 5 μM retinoic acid for additional 4 days.

### Cell line generation

cDNA encoding DNMT3A1 was amplified from ESC mRNA extracts and cloned into pL1-CAGGS-bio-MCS-polyA-1L or pCAGGS-bio-

MCS-IRES-BlasticidinR-polyA. Detailed cloning strategies and constructs are available upon request. Biotin-tagged DNMT3A1 cell lines were obtained by recombinase-mediated cassette exchange (RMCE) or by random integration as previously described (Baubec *et al*, 2013). *Dnmt3a*-KO cell lines were generated by transfecting px330 (Cong *et al*, 2013) with two guides targeting exon seven. Guide 1: CTCATACTCAGGCTCATCGT and guide 2: GTCGGAGAAG CAGGGTCCGT.

### Nuclear extract preparation for targeted proteomics and biotin co-immunoprecipitation MS

Cell pellets were resuspended gently in 4–5× pellet volume (PV) of nuclear extract buffer 1 (NEB1) with 10 mM HEPES pH 7.5, 10 mM KCl, 1 mM EDTA, 1.5 mM MgCl$_2$, 1 mM dithiothreitol, and 1× protease inhibitor cocktail and incubated for 10 min on ice. Cells were centrifuged at $2,000 \times g$ at 4°C for 10 min. Supernatant was removed and cell pellets were resuspended in 2× PV of NEB1 and homogenized. Suspension was centrifuged at $2,000 \times g$ at 4°C for 10 min twice and the pellet was resuspended in 1× PV and incubated with 300 U Benzonase per 1 ml total volume for 3 h at 4°C rotating, followed by centrifugation at $2,000 \times g$ at 4°C for 10 min. The pellet was resuspended in 1× PV of NEB2 (20 mM HEPES pH 7.5, 20% glycerol, 0.2 mM EDTA, 1.5 mM MgCl$_2$, 1 mM dithiothreitol, 1× protease inhibitor cocktail, and 450 mM NaCl) and homogenized. Nuclear proteins were extracted for 1 h at 4°C, and protein concentrations were measured by Qubit protein assays (TS).

For biotin co-immunoprecipitation, equal amounts of nuclear extracts (NE) were diluted with 2× the volume of 20 mM HEPES pH 7.5, 20% glycerol, 1.5 mM MgCl$_2$, 0.2 mM EDTA, 1 mM DTT, and 1× protease inhibitor cocktail and NP-40 was added to a final concentration of 0.3%. NE were incubated with 40 μl Streptavidin-M280 (11206D) beads overnight, rotating at 4°C. Beads were separated from the NE on a magnetic rack and washed once for 10 min rotating overhead at 4°C with 20 mM HEPES pH 7.5, 20% glycerol, 1.5 mM MgCl$_2$, 0.2 mM EDTA, 150 mM NaCl, 0.3% NP-40, 1 mM DTT, and 1× protease inhibitor cocktail. Beads were washed again with the same buffer without NP-40, and 10 min with 50 mM Tris pH 8.5, 1 mM EGTA, 75 mM KCl, each 10 min rotating overhead at 4°C. Proteins were eluted by adding 200 μl of 8 M urea (GEPURE00-66) in 100 mM Tris–HCl pH 8.2 and DTT at a final concentration of 0.1 M by shaking for 30 min at room temperature in a thermomixer. Separated supernatant was processed immediately for FASP digest (Wisniewski *et al*, 2009) using the whole eluate. Samples were then desalted with C18 ZipTips (ZTC18S960) according to the user manual.

### Mass spectrometry detection of DNMT3A-interacting proteins

Dissolved samples were injected by a Waters M-class UPLC system (Waters AG) and separated on a C18 reverse-phase column (HSS T3 1.8 μm, 75 μm × 250 mm, Waters AG). The column was equilibrated with 99% solvent A (0.1% formic acid (FA) in water) and 2% solvent B (0.1% FA in ACN). Peptides were eluted using the following gradient: 2–25% B in 50 min; 25–35% B in 10 min; 35–98% B in 5 min. The flow rate was constant, 0.3 μl/min. High accuracy mass spectra were acquired with a Q-Exactive HF mass spectrometer (Thermo Scientific) that was operated in

data-dependent acquisition mode. A survey scan was followed by up to 12 MS2 scans. The survey scan was recorded using quadrupole transmission in the mass range of 350–1,500 m/z with an AGC target of 3E6, a resolution of 120,000 at 200 m/z, and a maximum injection time of 50 ms. All fragment mass spectra were recorded with a resolution of 30,000 at 200 m/z, an AGC target value of 1E5, and a maximum injection time of 50 ms. The normalized collision energy was set to 28%. Protein identification and reporting from MS-raw data was performed with Mascot and Scaffold, respectively. Only proteins with a minimum of four unique spectral counts were utilized for the final dataset and contaminants were manually removed.

### Chromatin immunoprecipitation (ChIP)

For cross-linking and chromatin extraction, $10$–$20 \times 10^6$ cells were fixed for 8 min with 1% formaldehyde at room temperature followed by the addition of glycine (final concentration 0.12 M) and incubation for 10 min on ice. Cells were harvested and incubated for 10 min in 5 ml 10 mM EDTA, 10 mM TRIS, 0.5 mM EGTA on ice, followed by centrifugation at $680 \times g$ for 5 min. Cells were resuspended in 5 ml buffer containing 0.25% Triton X-100, 1 mM EDTA, 10 mM TRIS, 0.5 mM EGTA, and 200 mM NaCl and incubated for 10 min on ice followed by centrifugation at $680 \times g$ for 5 min. Final cell lysis was performed with 50 mM HEPES, 1 mM EDTA, 1% Triton X-100, 0.1% deoxycholate, 0.2% SDS, and 300 mM NaCl (medium salt buffer) in a pellet size-dependent volume (ranging from 0.9 to 1.2 ml) for 1 to 2 h on ice. Cross-linked chromatin was subjected to sonication in a Bioruptor Pico instrument (Diagenode) according to the manufacturer's instructions. Sonicated chromatin was centrifuged at $12,000 \times g$ for 10 min at 4°C and supernatant was used for further steps. Streptavidin-M280 magnetic beads were blocked for 1 h with 1% cold fish skin gelatin (Sigma Aldrich) and 100 ng tRNA (Sigma Aldrich) supplemented with protease inhibitor cocktail mix (Roche) and washed twice with buffer 3 with 0.1% SDS and 150 mM NaCl. 150–250 µg chromatin solution was diluted to 0.1% SDS and 150 mM NaCl. Chromatin was then incubated with 30 µl pre-blocked streptavidin-M280 magnetic beads overnight at 4°C. Beads were washed under rotation for 8 min for each wash step and placed on a magnetic rack for 2 min for exchange of buffers first with two rounds of 2% SDS, high salt buffer (as medium salt buffer, but 0.1% SDS and 500 mM NaCl), DOC buffer (250 mM LiCl, 0.5% NP-40, 0.5% deoxycholate, 1 mM EDTA, 10 mM TRIS), and two rounds of Tris/EDTA buffer. Beads were treated with RNaseA (60 µg, Roche) for 30 min at 37°C in 1% SDS, 0.1 M NaHCO₃, and subsequently proteinase K (60 µg, Roche) for 3 h at 55°C in 1% SDS, 0.1 M NaHCO₃, 10 mM EDTA, 20 mM TRIS, followed by de-cross-linking overnight at 65°C. DNA was purified with phenol–chloroform extraction and ethanol precipitation.

### 5-hydroxy-MeDIP-seq libraries

Genomic DNA was sonicated to 150–300 bp and RNaseA treated prior to end-repair and A-tailing using the NEB Ultra library kit and protocol (E7370). NEB adapters were ligated to the genomic DNA following the NEB Ultra protocol, and DNA was cleaned up using Ampure XP beads (Beckman Coulter). Following cleanup, 5 µg

adapter-ligated DNA was taken up in 500 µl TRIS/EDTA and denatured for 10 min in a boiling water bath. 5-hmC-containing DNA was enriched using the Active Motif 5-hmC antibody (39769) as described in Feldmann *et al* (2013). Enriched DNA was amplified by PCR with 15 cycles prior to sequencing.

### ChIP-seq library preparation and high-throughput sequencing

For ChIP-seq, libraries were prepared using the NEB-next ChIP-seq library Kit (E62402) following the standard protocols. Four to five samples with different index barcodes were combined at equal molar ratios and sequenced as pools. Sequencing of library pools was performed on Illumina HiSeq 2500 machines according to Illumina standards, with 75- to 150-bp single-end sequencing. Library demultiplexing was performed following Illumina standards.

### Reduced-representation bisulfite library preparation

Prior to library preparation, unmethylated lambda phage DNA and *SssI*-methylated T7 DNA were spiked in to genomic DNA samples to control bisulfite conversion rates. For RRBS, 20 µg of DNA was digested for 18 h by *MspI* and separated on 2% agarose gels. Digested DNA was excised at 150–250 and 250–350 bp. Digested DNA samples were end-repaired and A-tailed using the NEB Ultra Illumina library preparation kit (E7370). DNA fragments were ligated to methylated adapters (NEB E7535) following the NEB Ultra protocol. After adapter removal using Ampure XP beads (Beckman Coulter), DNA was converted using the Qiagen Epitect bisulfite conversion kit following the FFT sample protocol. After conversion, library PCR amplification was performed with 10 cycles following NEB Ultra kit protocol and cleaned up using Ampure XP beads. Two libraries were generated each for 150- to 250-bp and 250- to 350-bp inserts.

### Whole-genome bisulfite library preparation

Similar to RRBS, unmethylated lambda phage DNA and SssI-methylated T7 DNA were spiked in to the genomic DNA samples. For WGBS, 20 µg of DNA was sonicated to *ca* 500 bp using a Bioruptor Pico. Sonicated DNA were end-repaired and A-tailed using the NEB Ultra Illumina library preparation kit (E7370). DNA fragments were ligated to methylated adapters (NEB E7535) following the NEB Ultra protocol. After adapter removal using Ampure XP beads (Beckman Coulter), DNA was converted using the Qiagen Epitect bisulfite conversion kit following the FFT sample protocol. After conversion, library PCR amplification was performed with 10 cycles following NEB Ultra kit protocol and cleaned up using Ampure XP beads.

### ChIP-seq, 5-hMeDIP, and bisulfite sequencing reads processing

Samples were filtered for low-quality reads and adaptor sequences and mapped to the mouse genome (version mm9) using the BOWTIE algorithm allowing for two mismatches and only unique mappers were used. Identical reads from PCR duplicates were filtered out. For whole-genome bisulfite sequencing analysis, reads were partitioned into 40 bp and pooled for subsequent alignment. Bisulfite alignments were performed using QuasR in R (Gaidatzis

et al, 2015) with standard parameters for single-read bisulfite alignments. Methylation calls and read coverage per CpG were extracted and only CpGs covered more than ten times were used for subsequent analysis. CpGs overlapping with SNPs were removed. Bisulfite conversion quality was confirmed by spiked-in controls of methylated T7 DNA and unmethylated lambda DNA.

### Genomic coordinates

Genomic annotations are based on the *Mus musculus* version NCBI37/mm9 from July 2007. Genome segmentations based on DNA methylation corresponding to FMR, LMR, and UMR were obtained from Stadler *et al* (2011). For average density plots in Fig 1, UMRs and LMRs were filtered based on distance (> 6 kb) and FMRs larger than 12 kb were used to avoid overlaps between these features. Furthermore, FMRs were binned into three equal bins based on DNA methylation density. For identification and visualization of promoter UMRs, only UMRs were used that overlapped with gene promoters and these UMRs were oriented according to the gene orientation. UMR borders were defined at the end and start of the preceding and following FMR, respectively. CpG island annotation is based on the CpG cluster algorithm (Hackenberg *et al*, 2006). We removed all CpG islands that were shorter than 200 bp and retained all CpG islands overlapping with gene promoters. Promoter CpG islands with m-CpG average below 20% were assigned as unmethylated and further binned as bivalent based on H3K27me3 and H3K4me3 enrichments (Appendix Fig S4C). RefSeq genes and promoters were obtained from UCSC (genome.ucsc.edu), and only non-overlapping genes were used for further analysis.

### DNA methylation analysis

DNA methylation percentage, CpG and m-CpG densities per analyzed regions were calculated as described previously (Stadler *et al*, 2011; Baubec *et al*, 2013). Densities were calculated as the sum of SNP-filtered CpGs per analyses region and normalized to 100 bp. Global re-methylation was calculated as percent of methylated cytosines within different sequence contexts for the entire library. For WGBS, only CpGs covered more than ten times were considered; for RRBS, only CpGs covered more than 20 times.

### Genome-wide enrichments and identification of DNMT-enriched regions

The mouse genome was partitioned into 1-kb-sized tiling windows. Genomic regions overlapping with satellite repeats (Repeatmasker), ENCODE black-listed (ENCODE Project Consortium, 2012), and low mappability scores [(Derrien *et al*, 2012), below 0.5] were removed in order to reduce false positives arising from differences between mouse strains, annotation errors, and repetitiveness. Furthermore, DNA methylation scores in ES and NP cells windows with insufficient coverage in whole-genome bisulfite sequencing (> 50% of all CpGs covered at least 10 times) were removed. ChIP enrichments were calculated as $\log_2$-fold changes over input chromatin after library size normalization and using a constant of eight pseudo-counts to reduce sampling noise. ChIP enrichments were subsequently used for cross-correlation analysis, enrichment calculations

at DNMT-bound sites, or dynamic analysis in ES and NP cells. We utilized DEseq2 (Love *et al*, 2014) to identify regions exclusively bound by DNMT3 proteins. Prior to identifying these regions, we removed all genomic windows that did not contain sequencing read counts in any of the foreground or background samples and identified bound regions as FC > 1 and adjusted *P*-value < 0.002. Enriched regions from all three DNMT proteins were intersected and only regions exclusively bound by the individual DNMTs were further used to calculate distributions of genomic and epigenomic features, calculate DNA methylation, or generate meta-profiles using genomation in R (Akalin *et al*, 2014).

### Data access

All datasets produced in this study (Appendix Table S1) have been deposited to the NCBI Gene Expression Omnibus under the accession: GSE96529. Additional datasets from other publications are listed in Appendix Table S2.

**Expanded View** for this article is available online.

### Acknowledgements

We would like to thank Dirk Schübeler (FMI, Basel) for sharing cell lines and members of the DMMD (UZH) for sharing reagents. We thank Catharine Aquino and members of the Functional Genomics Center Zurich for high-throughput sequencing and mass spectrometry support, and the Science IT (S3IT) team at the University of Zurich for providing the computational infrastructure. Furthermore, we thank the anonymous reviewers, Oliver Bell (IMBA, Vienna), Dirk Schübeler (FMI, Basel), and members of the Baubec Lab for their critical input on the manuscript. Research in the Baubec Lab is supported by the Swiss National Science Foundation (P3_157488), the Swiss initiative in Systems Biology (SystemsX.ch), The Novartis Research Foundation (16A032), the Foundation for Research in Science and the Humanities at the UZH (STWF-16-008), the Alumni Association of UZH (ZUNIV), and the University of Zurich.

### Author contributions

Performed experiments: MM, JW, CA, RV, TB; Provided methodology and resources: RV and BR; Analyzed data: MM, JW, BR, TB; Designed study, supervised experiments, and wrote manuscript: TB.

### Conflict of interest

The authors declare that they have no conflict of interest.

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
