## [Review Process File · The EMBO Journal]

Manuscript EMBO-2017-97038

Isoform-specific localization of DNMT3A regulates DNA methylation fidelity at bivalent CpG islands

Massimiliano Manzo, Joël Wirz, Christina Ambrosi, Rodrigo Villaseñor, Bernd Roschitzki & Tuncay Baubec

Corresponding author: Tuncay Baubec, University of Zurich

Review timeline:

Submission date:	29 March 2017
Editorial Decision:	24 May 2017
Revision received:	22 August 2017
Editorial Decision:	25 September 2017
Revision received:	01 October 2017
Accepted:	03 October 2017

Editor: Anne Nielsen

Transaction Report:

1st Editorial Decision

24 May 2017

Thank you for submitting your manuscript to The EMBO journal and my apologies again that the review period lasted longer than expected. We have now finally heard back from all three referees and their comments are shown below.

As you will see from the reports, our referees praise the impact and quality of your study, although they also raise a number of concerns that you will have to address before they can support publication of the manuscript.

For the revised manuscript I would particularly ask you to focus your efforts on the following points:

- > Clarify how biotin-tagged DNMT3A1/2 behaves relative to the endogenous versions (expression, localization) as this point is raised by all three referees
- > Elaborate and discuss the data presentation and analysis to address the many minor comments from the refs
- > In addition, you will see that each referee points to different further reaching points that they would like to see in the study: ref #1 asks for insight on the differential DNMT3A1/2 recruitment, ref #2 wants H3K27me3 ChIPseq in TKO and re-expression lines, while ref #3 suggests using CRISPR to tag endogenous DNMT3A1/2. I realise that pursuing all these directions will be a lot of work and go beyond the scope of the current study and I would therefore be happy to discuss what kind of data you could potentially include in response to these points.

Given the referees' positive recommendations, I would like to invite you to submit a revised version of the manuscript, addressing the comments of all three reviewers. I should add that it is EMBO

Journal policy to allow only a single round of revision, and acceptance of your manuscript will therefore depend on the completeness of your responses in this revised version.

When preparing your letter of response to the referees' comments, please bear in mind that this will form part of the Review Process File, and will therefore be available online to the community. For more details on our Transparent Editorial Process, please visit our website: http://emboj.embopress.org/about#Transparent_Process

Thank you for the opportunity to consider your work for publication. I look forward to your revision.

REFEREE REPORTS

Referee #1:

The DNMT3A de novo DNA methyltransferase exists in two main forms -DNMT3A1 and DNMT3A2- that differ by the exclusion of the most N terminal part in the short DNMTA2 protein. Their relative expression is subject to developmental regulation; it is not known whether these two isoforms methylate the same genomic targets or not.

To address this question, the authors have used a method previously published by the PI for efficiently and precisely mapping DNMT binding sites using a bioID-derived approach in mouse ES cells and derivatives. Their conclusions are the following: while the short DNMT3A2 isoform covers up most of the genome, the long DNMT3A1 isoform shows greater specificity towards the edges of CpG islands (aka CpG island shores), which are occupied in their center by H3K27me3 marks. Interestingly, this targeting is seen even in absence of pre-existing DNA methylation, as observed upon expression of DNMT3A1 in DNA methylation-free ES cells. DNA methylation at the shore sites are associated with local TET-dependent hydroxymethylation, and indicate a dynamic turnover of DNA methylation. DNMT3A1 may then be required in turn for preventing hypomethylation and spreading of H3K27me3 into the CpG island shores. The performed experiments and analyses are usually well defined and solid, the conclusions are novel and appealing for a large audience of experts in developmental biology, epigenetic regulation and cancer, and the manuscript is well written from the introduction to the conclusion.

However, there are a few points I would like to see addressed, as listed below. Notably, these relate to both potential non-specificity due to overexpression assays and the cues that may explain the specific targeting of DNMT3A1 versus DNMTA2 and DNMT3B1.

Main points:

1- The authors should mention previous works that reported the differential subcellular localization of DNMT3A1 and DNMT3A2. Also, what is known about their relative catalytic efficiency in biochemical assays?

2- One important piece of information is the relative transcript level of the endogenous DNMTA1 and A2 isoforms in ES cells and NPCs (which are the two cellular models onto which experiments were performed). Could the authors provide this quantification? From published CAGE-seq datasets or by targeted RT-qPCR?

3- DNMT3A1 is actually known to be much more lowly expressed than DNMT3A2 in mES cells. What is the relative expression level of the biotin-tagged DNMT3A1 transgene compared to the endogenous DNMT3A1 isoform at the protein level? If we are in non-physiological conditions of

strong DNMT3A1 overexpression, this should be specified in the text. On the same tone, although the different tagged DNMT3 transgenes are located at the same position by RMCE, how do their expression levels compare to each other? Similar expression levels would be required when comparing DNMT3A1 and DNMT3A2 binding distribution.

4-ES cells were cultured in high methylation-promoting conditions (serum-based), therefore the breadth of de novo DNA methylation that is gained upon NPC differentiation is quite limited. The authors chose to focus their efforts on the few regulatory regions that gain methylation in this differentiation system. Could there be a bias in selecting only on these regions? These are somehow unusual compared to the rest of the genome that is fully methylated in both serum-grown ES and NPCs. I do not have any specific experiments in mind to resolve this question but it would be informative to how many regions it represents (how many regions gain DNA methylation from serum-ES cells to NPCs and rely on DNMT3A1).

5- Why does it matter to have DNA methylated versus hypomethylated/H3K27me3 methylated CpG island shores? Is there any impact on expression? In this regard, it would be very informative to have the comparative RNA-seq analysis of Dnmt-tKO vs Dnmt-tKO+DNMT3A1 cells (provided that there is enough DNA methylation in the absence of DNMT1?).

6- Finally, the burning question: why DNMT3A1 has this specific preference towards the shores of polycomb-regulated CpG islands? Is the N-terminal part -which is missing in DNMT3A2 and highly divergent in DNMT3B- required for this localization? It seems essential to me to repeat the bioChIP with a transgene carrying only the DNMT3A1-specific N-terminal part and see whether this domain is sufficient or not for DNMT3A1 localization.

Minor points:

- Fig 1d: I guess that E12, E14 etc.. refer to "Embryonic Days"? Please provide information in the legend.
- genomic coordinates for screen shots in Fig 3c?
- Fig 4a: what is the difference between the two heat map panels for DNMT3A1 binding sites?

Referee #2:

Manzo et al report the identification of distinct roles for the different isoforms of the de novo DNA methyltransferase DNMT3A in mouse cells. The authors analyze mouse ES cell lines in which the DNMT3A1 isoform has been tagged with a peptide that gets biotinylated, allowing its enrichment by ChIP to analyze its localization genome-wide.

They report isoform specific binding patterns for DNMT3A1 and DNMT3A2, and in particular the enrichment of DNMT3A1 at bivalent (H3K4me3 + H3K27me3) CpG island promoters.

Furthermore, while both DNMT3A isoforms were associated with de novo methylated sites during ES cell differentiation into neuronal progenitor cells, differential DNMT3A1 binding is uniquely associated with regions exhibiting dynamic H3K27me3 during differentiation, with a positive correlation between DNMT3A1 and H3K27me3 enrichment observed, and frequent associated changes in DNA methylation level. The authors observe that polycomb-associated promoter CpG islands that display H3K27me3 exhibit enrichment of both DNMT3A1 binding and 5hmC at the CpG island borders, and that DNMT3A1 localization and activity both forms the substrate for TET mediated formation of 5hmC, as well as counteracting the spreading of the unmethylated state into surrounding regions, as supported by CpG island border loss of DNA methylation and 5hmC in Dnmt3a knock-out lines. Introduction of DNMT3A1 into a DNMT triple knockout line resulted in increased methylation at DNMT3A1-enriched regions and bivalent CpG islands.

Altogether, this is an interesting study that is well written and mostly well presented (see comments). This work provides new insights into the roles of DNMT isoforms and their roles at bivalent CpG islands, where relatively limited exploration of isoform specific roles has been undertaken in the past. The novelty of the manuscript lies in dissecting the distinct genomic

localization and roles of the DNMT3A isoforms, and identifying links between DNMT3A1 and DNA methylation, 5hmC, and H3K27me3 at bivalent CpG islands. Together, these results suggest that DNMT3A1 may function to counteract the spreading of an unmethylated state into surrounding DNA, as well as constrain the spread of H3K27me3 at these promoters.

Comments:

- Is it possible that the N-terminal tag added in the RAMBiO method could affect the localization of each isoform differently? e.g. can the possibility be excluded that the tag disrupts DNMT3A2 structure, interactions or function in a manner that alters its genomic localization, leading to the apparent isoform specific localization of DNMT3A1 and DNMT3A2? Does usage of a DNMT3A antibody that recognizes both isoforms identify the union of DNMT3A1 and DNMT3A2 bound regions identified in this study?

- From Fig 1d and S1i it does not appear that the ChIP enrichment level is particularly high. It's not clear that the binding signal is highly enriched in the regions depicted in these figures, making the stated point of the locus specific binding of the DNMT3A isoforms somewhat challenging to discern. Do other loci show more clear IP enrichment and differences between the DNMT3a isoforms?

- Is the apparent depletion of mC flanking DNMT3A1 binding sites an artefact of combining binding sites at the 5' and 3' of CGIs, i.e. If CGI 5' binding sites were separated from 3' binding sites, would the depletion look symmetrical or directional?

- Fig 2e, have CpG island promoters / TSSs been oriented so that all are represented at the 5' of genic regions, i.e. in the same direction as the associated gene? The symmetry of the H3K36me3 signal around the promoter CpG islands would suggest that CpG islands on the Crick strand have not been reversed, compared to the H3K36me3 asymmetry observed in Fig 4a. If not already done, it would be useful to orient all in the same direction relative to the gene coding direction so that any asymmetric features can be discerned.

- It can be challenging to interpret how broadly a pattern is observed when only metaplots are used that aggregate signal over all members of a set of genomic regions. For example, from Fig 5a it is not clear how broadly the reintroduction of DNMT3A1 into the TKO cells restores DNA methylation at DNMT3A1 sites or bivalent CpG islands. Are all regions restored to the same degree, or is there heterogeneity, and if so, is it related to any other features? It would be very helpful to also include heatmap representations of the signal at these regions, as done in Fig 4a, even if only included in the supplement due to space restrictions, so that this could be explored further. This would also benefit other analyses e.g. those presented in Fig 4e, 5b, 5d.

- ChIP-seq of H3K27me3 should be performed in TKO, TKO + DNMT3A1, and TKO + DNMT3A2 cells in order to determine in this system whether loss of DNA methylation results in spreading of H3K27me3 and central depletion, and whether this spreading would be limited or prevented upon restoration of DNMT3A1 activity, but not DNMT3A2.

- The increased 5-hmC production around bivalent CpG islands is not clear from Fig 5c. This should be clarified in the figure, or genomic regions that more clearly exemplify these patterns should be presented.

- More details about library characteristics and quality should be provided in the supplement, such as number of reads, mapping rate, coverage (where appropriate), and bisulfite conversion quality/frequency.

- Fig 1c, should show DNMT3a2 binding over genes too

Referee #3:

DNA modification at cytosine residues is a prevalent feature of vertebrate genomes and its presence at regulatory elements can constitute an additional layer, in concert with gene regulatory networks (GRNs), for control of gene expression. This represents an entry point for DNA modification in processes such as X inactivation, repeat sequence inactivation, imprinting, and differentiation. We also know that DNA methylation landscapes impacts on other chromatin modifying activities and can thus influence gene expression states indirectly.

The DNA methylation machinery in mouse embryonic stem cells (mESCs) is intriguing as it has multiple components; multiple de novo methyltransferase isoforms, co-factors (Dnmt3l, UHRF1, Sirt1) and methylcytosine oxidases (Tet enzymes) that maintain a dynamic pattern of methylation. Yet fully hypomethylated mESCs are viable. Here Manzo and colleagues report that a specific de novo isoform, DNMT3A1 preferentially localises to the methylated shores of bivalent CpG islands (CGI), whereas its shorter isoform DNMT3A2 is globally distributed throughout the genome. They suggest that DNMT3A1 is required to protect CpG island shores from hypomethylation by counteracting TET-mediated oxidation of methylated cytosine. For the latter point, it could be equally argued that TET-mediated oxidation of methylated cytosine protects CpG island shores from hypermethylation. This illustrates that it is difficult to draw substantive conclusions from the experiments presented.

I think the data presented is of a high quality and expertly analysed to an excellent standard. The authors should be congratulated on this aspect, but I have some comments that need to be addressed.

(A) Experimental:

1. The experiments essentially depend on an over expression system of tagged proteins based on RCME insertion of constructs driven from the CAGGs promoter in mESCs and derived cells. What is the evidence that this system mimics endogenous expression of the particular isoforms and the resulting 5mC patterns? A long and considered answer is expected for this question.
2. Does over-expression of the particular isoform under test unbalance the system? For example is the stoichiometry between co-factors (e.g. Dnmt3l) and other partners (other Dnmt3s) when massive amounts of the exogenous protein are present, may this affect the DNA methylation pattern outcomes?
3. Does co-expression of isoforms (A1, A2, 3B or 3L) essentially give the same result as single gene over-expression?
4. One possible way to address these questions (2 and 3) is to use CRISPr technologies to tag endogenous genes (either with the Bio or another tag) and validate if the ChIP patterns match the over-expression results.

(B) The manuscript and experimental:

The authors state that an illustration on how important DNMT3A methylation is, is the phenotypes of mice with mutant DNA methyltransferases. However, this does not square with viable hypomethylated mice resulting from mutations in co-factors such as HELLs (very hypomethylated) or UHRF1 (10% reduction in global methylation). The authors should adjust the introduction accordingly as the phenotypes of hypomethylated mice depend on their derivation route. Given their conclusions, it should be noted that DNMT3A KO mice are sub-viable; they get through early development fine. DNMT3A^{-/-} mice develop to term and appear to be normal at birth. However, most homozygous mutant mice become runted and die at about 4 weeks of age, long after the potential events described in this report.

The role of the new isoform, DNMT3C, should be mentioned in the introduction, what are their expectations of how this DNMT3 isoform will perform in their system?

Primary citations should be used for the statement that 'DNA methylation is highly dynamic and undergoes constant turnover at regulatory sites'. I suggest that the following are very appropriate: PMID 26928226 and 27346350.

The statement 'H3K9me3 plays a role in maintenance of DNA methylation via the accessory protein UHRF1 (Rothbart et al, 2012; Meyenn et al, 2016)' is erroneous because Meyenn links H3K9me2 with maintenance methylation and this can be challenged by PMID 27554592, which concludes that 'while our study (with a Uhrf1 knockin (KI) mouse model) supports a role for H3K9 methylation in

promoting DNA methylation, it demonstrates for the first time that DNA maintenance methylation in mammals is largely independent of H3K9 methylation.'

The last paragraph in the introduction appears to be a reproduction of the abstract; can they be a bit more informative?

Regarding the CAGE data, can the authors expand on this to include DNMT3C, DNMT1S, DNMT1O and DNMT3L. With respect to their proposal regarding 5hmC, a TET isoform CAGE survey would also be informative. Their survey should include as many stages (including adult) as possible and tissue types.

The IHC in Figure S1D is of poor resolution and lacks sizing bars, a better picture would be appreciated.

While the 5hmC argument is interesting, it is not compelling as it appears correlative and not causative. What happens if Tets are inactivated in the DNMT3A1 over-expressing mESCs, do you get methylation creep at the coastal regions of the CGIs?

Figure 4a requires a 5mC heat-map profile in addition to 5hmC.

During NPC derivation, how does expression of the endogenous DNA methylation machinery change (DNMT-1,-3A,-3B,-3C and 3L1; Tet1, 2 and 3), how will this impact on the outcome of over-expressing DNMT3 isoforms in this system compared to mESCs?

The data in figure 5c-d, must be interrogated in the context of the recent publication by Xiong et al (2016: PMID-27840027), who suggest that there is collaborative interaction between SALL4A and TET proteins in regulating stepwise oxidation of 5mC at enhancers. What is the profile of enhancer 5hmC in the TKO-DNMT3A1 and TKO-DNMT3A2 rescue cell lines? Is this restored without requirement for DNMT1?

I was bit surprised that Arnand et al (2012: PMID- 22761581) was not discussed, who identified a substantial incomplete regional methylation maintenance and importantly that non-CpG cytosine methylation is confined to ESCs and exclusively catalysed by DNMT3A and DNMT3B, is there any difference in non-CpG methylation in their various rescued cell lines? Is non-CpG methylation accentuated in the transgenic mESCs and NPCs?

Response to Reviewer's comments

We very much appreciate the opportunity to submit a revised version of our manuscript "**Isoform-specific localization of DNMT3A regulates DNA methylation fidelity at bivalent CpG islands**". We are excited about the very encouraging comments from the reviewers, who acknowledged the significance of our findings, and we thank them for their constructive criticism to our first submission and for suggesting additional experiments that helped us to improve the manuscript.

As you will see from our responses and in the revised manuscript, we have now clarified the expression levels of the biotin-tagged DNMT3A1/2 isoforms in ES cells by applying quantitative targeted mass spectrometry to measure the protein levels of DNMT3A in the engineered cells, revealing a moderate increase in total DNMT3A levels. To exclude that this moderate increase influences the genomic binding of DNMTs, we make use of clones with lower expression of the transgenes and show that protein levels do not influence their genome-wide localization.

In addition, as part of our attempts to understand how isoform-specific localization is regulated, we provide now protein-protein interaction analysis by mass spectrometry for both isoforms. Most importantly, through expressing a chimeric DNMT3A protein that carries the N-terminal end of DNMT3B *in vivo*, we show that the DNMT3A1-specific N-terminal part is essential for the observed localization to H3K27me3 sites.

By extending our computational analysis and by including additional datasets we further substantiate that the sites preferentially bound by DNMT3A1 have an elevated turnover of DNA methylation and coincide with the H3K27me3-decorated promoters of relevant transcription factors which are frequently deregulated in absence of DNA methylation.

These additions provide further support for our main finding that DNMT3A1 is targeted to Polycomb-regulated promoters to regulate DNA methylation turnover at CpG island shores. These new experiments led to 1 new main figure and 7 new Supplementary figures. Below we respond point-by-point to the comments of each reviewer.

Referee #1:

The DNMT3A de novo DNA methyltransferase exists in two main forms -DNMT3A1 and DNMT3A2- that differ by the exclusion of the most N terminal part in the short DNMTA2 protein. Their relative expression is subject to developmental regulation; it is not known whether these two isoforms methylate the same genomic targets or not.

To address this question, the authors have used a method previously published by the PI for efficiently and precisely mapping DNMT binding sites using a bioID-derived approach in mouse ES cells and derivatives. Their conclusions are the following: while the short DNMT3A2 isoform covers up most of the genome, the long DNMT3A1 isoform shows greater specificity towards the edges of CpG islands (aka CpG island shores), which are occupied in their center by H3K27me3 marks. Interestingly, this targeting is seen even in absence of pre-existing DNA methylation, as observed upon expression of DNMT3A1 in DNA methylation-free ES cells. DNA methylation at the shore sites are associated with local TET-dependent hydroxymethylation, and indicate a dynamic turnover of DNA methylation. DNMT3A1 may then be required in turn for preventing hypomethylation and spreading of H3K27me3 into the CpG island shores. The performed experiments and analyses are usually well defined and solid, the conclusions are novel and appealing for a large audience of experts in developmental biology, epigenetic regulation and cancer, and the manuscript is well written from the introduction to the conclusion.

We thank the reviewer for this positive comment.

However, there are a few points I would like to see addressed, as listed below. Notably, these relate to both potential non-specificity due to overexpression assays and the cues that may explain the specific targeting of DNMT3A1 versus DNMTA2 and DNMT3B1.

Main points:

1- The authors should mention previous works that reported the differential subcellular localization of DNMT3A1 and DNMT3A2. Also, what is known about their relative catalytic efficiency in biochemical assays?

Response 1. We have now have included Chen et al 2002 (PMID 12138111), Chen et al 2003 (PMID 12897133) and Choi et al 2011 (PMID 20841325) as a reference for subcellular localization, catalytic efficiency and in vitro targeting specificity of the DNMT3A isoforms.

2- One important piece of information is the relative transcript level of the endogenous DNMT3A1 and A2 isoforms in ES cells and NPCs (which are the two cellular models onto which experiments were performed). Could the authors provide this quantification? From published CAGE-seq datasets or by targeted RT-qPCR?

Response 2. We now provide several measurements of DNMT3A isoform levels in these cell lines. We have performed RT-qPCR to measure isoform-specific transcript levels of DNMT3A in ES and during differentiation to neuronal progenitor cells (see new Figure S1B). Furthermore, we provide mRNA-seq tracks for ES and NPC cells to visualize transcription over the entire gene (see new Figure S1A). Finally, we have included immunoblot detection for both DNMT3A isoforms in ES and NPC cells using an antibody that detects both isoforms (see new Figures S2C and S9A). The FANTOM consortium does not provide CAGE-seq data from the *in vitro*-derived neuronal progenitor cells utilized in this study.

The new results indicate that both isoform transcripts are present in ES and NP cells (Figures S1A and S1B), and that both isoforms are higher expressed in NP cells. This is mirrored by the immunoblot results which indicate an increase of both isoforms in NP cells. However, the DNMT3A1 isoform is not detectable in ES cells by immunoblot. We think this is an antibody sensitivity issue since the longer isoform has been already detected in ES cells in previous studies (Chen et al., 2002; PMID 12138111).

3- DNMT3A1 is actually known to be much more lowly expressed than DNMT3A2 in mES cells. What is the relative expression level of the biotin-tagged DNMT3A1 transgene compared to the endogenous DNMT3A1 isoform at the protein level? If we are in non-physiological conditions of strong DNMT3A1 overexpression, this should be specified in the text. On the same tone, although the different tagged DNMT3 transgenes are located at the same position by RMCE, how do their expression levels compare to each other? Similar expression levels would be required when comparing DNMT3A1 and DNMT3A2 binding distribution.

Response 3. The reviewer raises the valid concern if our results are dependent on the expression level of the transgene, and that this could create non-physiological conditions that alter binding behaviour of the DNMT3A isoforms to the genome. In most of our experiments we have utilized the CAG promoter to drive expression of biotin-tagged DNMT proteins. The reason for using the CAG promoter is maintenance of stable expression from the RMCE site, i.e. expression does not decline after prolonged passaging or through differentiation as observed for other promoters - which is essential to study binding behaviour in ES cell derived progenitors. While this is indeed a robust promoter, we want to note that we express the transgene from a single locus on one allele only. Therefore, the obtained expression levels are not comparable to strong transgene "overexpression" levels observed in transient or random integration experiments which usually result in multiple copies of the same transgene expressed per cell.

Based on Immunoblot analysis (Figure S2C), the levels of the tagged isoforms in comparison to the endogenous proteins are indeed higher, by approximately 2 to 3-fold. However, based on this Immunoblot analysis we cannot make reliable statements about the protein levels. As part of our efforts to quantify this heterologous expression, and to test if expression level influences binding, we have now performed parallel reaction monitoring (PRM) mass spectrometry to measure DNMT3A protein levels in a targeted manner directly from nuclear extracts. This allowed us to accurately identify and quantify specific DNMT3A peptides from trypsin-digested nuclear extracts, and compare their abundance between wild type ES cells and cells expressing either biotin-tagged DNMT3A1 or DNMT3A2. For this we have used 4 different peptides that are shared between the DNMT3A isoforms, and measured them in four independent replicates each. These new results which we present now in Figure S2D and S2E indicate that the total abundance of DNMT3A (tagged and endogenous DNMT3A together) in these cell lines is increased by 2.6-fold in the DNMT3A1 clone_1 and by 2.5-fold in the DNMT3A2 cell line, when compared to wild type cells (1.4 and 1.29 in log₂-FC). We now mention this increase in expression in the manuscript (page 5). While this is indeed above the endogenous expression — we argue that the increase in protein levels is rather moderate and should not influence the results obtained in this study.

To finally test if this expression level has an influence on the observed binding of the DNMT3A isoforms, we have included data from a CMV-regulated biotin-tagged DNMT3A2 transgene that we have used in a previous publication to address a similar concern raised by the reviewers at this time (Baubec et al, 2015,

Supplemental Figure 1f and g — we have included the figure from this publication as Rebuttal Figure 1 below). As can be observed from the Immunoblot in the Rebuttal Figure 1, the CMV-driven DNMT3A2 protein is much lower expressed compared to CAG-DNMT3A2. Despite this difference in expression, we did not observe extensive differences in genomic binding for DNMT3A2. We have now included this dataset to our analysis where we compare the binding of the individual isoforms, and show that the correlation between identical isoforms remains high, independent of their expression levels (Figure S3B).

Furthermore, we have included two individual clones of biotin-tagged DNMT3A1. As can be noted from the Immunoblot in Figure S2C where we detect the biotin-tagged proteins using Streptavidin-HRP, the two individually-derived clones of DNMT3A1 show different expression levels, whereas clone 2 is expressed much lower than clone_1 (the higher-expressing clone_1 was also used to quantify DNMT3A levels in the PRM experiments above). Despite this difference in expression, we do not observe any influence on the genome-wide localization of DNMT3A1, as can be seen from numerous figures where we show either genome browser tracks for both clones or correlations between the two clones (Figures 1E and F, S3A,B,E and F).

Regarding the comparative expression levels of both isoforms from the RMCE site: The DNMT3A1 clone 1 shows similar expression levels as DNMT3A2, while DNMT3A1 clone_2 is expressed at lower levels (Figure S2B). In our analysis we matched expression levels of the proteins that we compare, however, since we do not observe any differences in binding upon weaker expression of DNMT3A1 or DNMT3A2, we suggest that this does not influence isoform-specific binding preferences.

We hope that this set of experiments, address the concerns of the reviewer sufficiently.

4-ES cells were cultured in high methylation-promoting conditions (serum-based), therefore the breadth of de novo DNA methylation that is gained upon NPC differentiation is quite limited. The authors chose to focus their efforts on the few regulatory regions that gain methylation in this differentiation system. Could there be a bias in selecting only on these regions? These are somehow unusual compared to the rest of the genome that is fully methylated in both serum-grown ES and NPCs. I do not have any specific experiments in mind to resolve this question but it would be informative to how many regions it represents (how many regions gain DNA methylation from serum-ES cells to NPCs and rely on DNMT3A1).

Response 4. We fully agree that the change in CpG methylation between serum-grown ES cells and NP is limited when calculated over the entire genome. However, previous publications have identified *de novo* DNA methylation at a number of regulatory regions using the same cellular differentiation system (Mohn et al 2008; Stadler et al 2011). The reported sites that gain DNA methylation include stage-specific proximal and distal regulatory elements, with 343 promoters (Mohn et al., 2008, PMID 18514006) and 22,184 ES-specific distal regions (ES LMRs) gaining *de novo* methylation in NPC (Stadler et al 2011, PMID 22170606).

We have now calculated the percentage of the genome that gains DNA methylation based on our 1kb window approach that is independent on the functional annotation of the underlying DNA sequences and provide this information in Figures S5A and S5B and corresponding figure legend, as requested by the reviewer. Taken together, the regions that display *de novo* methylation make up 1.4 % of the entire genome — this corresponds to 1.2 MB containing 23,725 CpGs.

For the presented results, it is important to note that our emphasis was on isoform-specific binding in correlation to changes in H3K27me3. Regions that gain H3K27me3 in NPs make up 8.1 % of the genome (Figure S5D). While we observe that some sites that gain DNMT3A1 binding based on H3K27me3 also become *de novo* methylated in NPs, the changes in H3K27me3 and *de novo* methylation are largely independent from each other - which we show now in Figure S5G.

Rebuttal Figure 1. From Baubec et al., 2015 - Supplemental Figure 1. f) Immunoblot -detection of DNMT3A2 in cell lines utilizing either a CMV or a CAG promoter to drive expression of the biotin-tagged transgene. g) Genome-wide correlation shows similar binding between DNMT3A2 expressed from CMV and CAG promoters. Shown are log₂-transformed read counts at 1-kb tiles covering the entire genome. Pearson's correlation is shown.

5- Why does it matter to have DNA methylated versus hypomethylated/H3K27me3 methylated CpG island shores? Is there any impact on expression? In this regard, it would be very informative to have the comparative RNA-seq analysis of Dnmt-tKO vs Dnmt-tKO+DNMT3A1 cells (provided that there is enough DNA methylation in the absence of DNMT1?).

Response 5. We have now re-analyzed RNA-seq and DHS-seq experiments performed in replicates and in isogenic WT and *Dnmt*-TKO cells lines (Domcke et al., 2015 PMID 26675734) to test if the unmethylated CpG island promoters we have identified to be preferentially enriched by DNMT3A1 are deregulated in absence of DNA methylation. We indeed observe that numerous DNMT3A1-target promoters show increased gene expression, as well as increased accessibility and H3K27ac in the *Dnmt*-TKO cells, suggesting that loss of DNA methylation at the shores and concomitant reduction of H3K27me3 would result in de-regulation of these genes (New Figure 6E and S12 D to G).

However, as anticipated by the reviewer, the “re-established” DNA methylation in TKO cells expressing DNMT3A1 or DNMT3A2 in absence of DNMT1 is not sufficient to reach DNA methylation levels similar to those observed in WT cells, and therefore unlikely to restore H3K27me3 levels and transcription. Based on our WGBS results at CpG dinucleotides shown in Sup Fig S10, only 8 % of CpGs are methylated in TKO-DNMT3A1 vs. 80% in WT cells. A recent publication from the Fang Lab further supports this conclusion (King et al 2016, PMID 27681438). Here the authors have reintroduced individual DNMT3 isoforms to *Dnmt*-TKO and *Dnmt3a/3b*-double-KO (DKO) cells and have measured H3K27me3 and other histone marks in the reconstituted cell lines. In general, their conclusion was that H3K27me3 and other marks can be partially restored by re-expressing *de novo* DNMTs, but this is only observed in the *Dnmt*-DKO cells, suggesting that maintenance by DNMT1 is required. In addition, their datasets — which we have re-analyzed in our study — strongly support that re-introduction of DNMT3A1 to *Dnmt*-DKO leads to resetting of H3K27me3 at the DNMT3A1-enriched Polycomb CpG islands identified in our study (Figures S12A to C).

6- Finally, the burning question: why DNMT3A1 has this specific preference towards the shores of polycomb-regulated CpG islands? Is the N-terminal part -which is missing in DNMT3A2 and highly divergent in DNMT3B- required for this localization? It seems essential to me to repeat the bioChIP with a transgene carrying only the DNMT3A1-specific N-terminal part and see whether this domain is sufficient or not for DNMT3A1 localization.

Response 6. We have now addressed this important question by generating new ES cell lines expressing a chimeric version of DNMT3A where we have replaced the N-terminal part of DNMT3A1 with the N-terminal part from DNMT3B. The reason for choosing this strategy was because i) this is the major part of the protein that varies between DNMT3A1 and DNMT3B, ii) we do not have any information about the structural properties of the DNMT3A1 N-terminal part in order to introduce rationally-designed mutations and iii) the N-terminal part alone — as suggested by the reviewer — leads to unspecific binding to accessible regions (see below). Utilizing the N3B-DNMT3A cell line, we have performed ChIP-seq to test if replacement of the N-terminal domain influences the genome-wide binding of DNMT3A. We observe that binding of this chimeric protein is reduced at CpG island shores of H3K27me3-positive CpG islands (new Figures 4C, D and S6), strongly supporting a role for the N-terminal part in specifying the targeting preference of DNMT3A1.

(Text related to figure for referees not shown)

Taken together, these results indicate that the N-terminal domain is required, but not sufficient for specifying the observed interactions of DNMT3A1 with H3K27me3 sites in the genome, which rather result from a combination of multiple domains (incl. ADD, PWWP and the catalytic domain). However, we wish not to include the dataset from the N-terminal domain alone in the manuscript, as we think that the obtained results are likely an artifact, and more detailed biochemical experiments are required

(Figure for referees not shown)

to resolve the individual contribution of this N-terminal part alone.

Minor points:

- Fig 1d: I guess that E12, E14 etc.. refer to "Embryonic Days"? Please provide information in the legend.
- genomic coordinates for screen shots in Fig 3c?
- Fig 4a: what is the difference between the two heat map panels for DNMT3A1 binding sites?

Response 7. We have added the requested information to the legend of Figure 1 and the coordinates for Figure 3D. In figure 4A, the two heat maps for DNMT3A1 represent independent bioChIPs from the two clones generated in this study, we have now omitted one of these replicates in order to accommodate the m-CpG heat maps requested by reviewer 2.

Referee #2:

Manzo et al report the identification of distinct roles for the different isoforms of the de novo DNA methyltransferase DNMT3A in mouse cells. The authors analyze mouse ES cell lines in which the DNMT3A1 isoform has been tagged with a peptide that gets biotinylated, allowing its enrichment by ChIP to analyze its localization genome-wide.

They report isoform specific binding patterns for DNMT3A1 and DNMT3A2, and in particular the enrichment of DNMT3A1 at bivalent (H3K4me3 + H3K27me3) CpG island promoters.

Furthermore, while both DNMT3A isoforms were associated with de novo methylated sites during ES cell differentiation into neuronal progenitor cells, differential DNMT3A1 binding is uniquely associated with regions exhibiting dynamic H3K27me3 during differentiation, with a positive correlation between DNMT3A1 and H3K27me3 enrichment observed, and frequent associated changes in DNA methylation level. The authors observe that polycomb-associated promoter CpG islands that display H3K27me3 exhibit enrichment of both DNMT3A1 binding and 5hmC at the CpG island borders, and that DNMT3A1 localization and activity both forms the substrate for TET mediated formation of 5hmC, as well as counteracting the spreading of the unmethylated state into surrounding regions, as supported by CpG island border loss of DNA methylation and 5hmC in Dnmt3a knock-out lines. Introduction of DNMT3A1 into a DNMT triple knockout line resulted in increased methylation at DNMT3A1-enriched regions and bivalent CpG islands.

Altogether, this is an interesting study that is well written and mostly well presented (see comments). This work provides new insights into the roles of DNMT isoforms and their roles at bivalent CpG islands, where relatively limited exploration of isoform specific roles has been undertaken in the past. The novelty of the manuscript lies in dissecting the distinct genomic localization and roles of the DNMT3A isoforms, and identifying links between DNMT3A1 and DNA methylation, 5hmC, and H3K27me3 at bivalent CpG islands. Together, these results suggest that DNMT3A1 may function to counteract the spreading of an unmethylated state into surrounding DNA, as well as constrain the spread of H3K27me3 at these promoters.

We thank the reviewer for acknowledging the relevance of our study.

Comments:

- Is it possible that the N-terminal tag added in the RAMBiO method could affect the localization of each isoform differently? e.g. can the possibility be excluded that the tag disrupts DNMT3A2 structure, interactions or function in a manner that alters its genomic localization, leading to the apparent isoform specific localization of DNMT3A1 and DNMT3A2? Does usage of a DNMT3A antibody that recognizes both isoforms identify the union of DNMT3A1 and DNMT3A2 bound regions identified in this study?

Response 8: The reason for choosing N-terminal over C-terminal tagging is to ensure that the tagged version is translated in full length and sufficiently biotinylated (see below). In addition, the biotin tag is 17 AA short and so far we have tested over 30 different proteins using the N-terminal tagging method. For many of

those we have compared the biotin-ChIP binding profiles to available antibody ChIPs resulting in reproducible maps and confirming the suitability of the N-terminal tagging strategy for genome-wide studies.

In case of DNMT3A, all antibodies we have tested so far (N=3) were not ChIP-seq grade and cross-reacted with numerous other proteins including DNMT3B on Immunoblots. Besides this, the low signal to noise ratios obtained from ChIP-seqs with such antibodies makes it almost impossible to discern bound regions from the background. This is also the primary reason why we avoided antibodies in our study. The high-stringency that can be applied to biotin-ChIP (2 % SDS and 500 mM NaCl/LiCl washing steps) allowed us to increase the signal to noise ratio, which was detrimental for detecting DNMT3 binding to the genome.

Nevertheless, we agree with the reviewer that we cannot rule out that the N-terminal tagging could influence DNMT3A2 interactions. We have attempted to introduce the tag to the C-terminus of DNMT3A1 and DNMT3A2 and we have generated the corresponding ES cell lines for this reason. However, while stable expression of the transgenic protein could be detected on Immunoblots using antibodies against DNMT3A (Rebuttal Figure 3, top blot), we observed that the proteins were not efficiently *in vivo* biotinylated (Rebuttal Figure 3, bottom blot) - when compared to the N-terminal tagged proteins used in this study (compare to Supplemental Figure S2C). We performed several bio-ChIPs with these clones, however we could not recover sufficient DNA material, resulting in flat signals in the ChIP-seq attempts.

Rebuttal Figure 3. Immunoblot detection of C-terminally tagged DNMT3A1 and DNMT3A2 isoforms expressed in ES cells. Shown are nuclear extracts from the indicated cell lines probed with an anti-DNMT3A antibody (top) and the same membrane re-probed with Streptavidin-HRP after blocking in 5% BSA (bottom). Both C-terminally tagged proteins are expressed and detected by the DNMT3A antibody (top). Only faint signals could be detected by Streptavidin (bottom, asterisks) indicating insufficient *in vivo* biotinylation of the introduced proteins. The endogenous biotinylated proteins serve as loading controls.

However, to test if the N-terminal tagging would lead to differential protein-protein interactions of the individual isoforms, which could influence their genomic localization, we performed now co-immunoprecipitation assays for both N-terminally tagged DNMT3A isoforms, followed by mass-spectrometric detection of the interacting proteins. By comparing the interactomes of DNMT3A1 with DNMT3A2 we observe that the interaction profiles of both isoforms are highly similar (Pearson's = 0.93) and both proteins enrich for known interactors, such as DNMT3L, G9A, DNMT3B, FACT and NuRD complex (new Figures 4A and B). These results, which to our knowledge are the first to compare protein-protein interactions of individual DNMT3A isoforms, suggest that the N-terminal tagging does not lead to differential interactions between the tested isoforms. We hope the reviewer agrees with this conclusion.

- From Fig 1d and S1i it does not appear that the ChIP enrichment level is particularly high. It's not clear that the binding signal is highly enriched in the regions depicted in these figures, making the stated point of the locus specific binding of the DNMT3A isoforms somewhat challenging to discern. Do other loci show more clear IP enrichment and differences between the DNMT3a isoforms?

Response 9: We agree that the DNMT3 protein ChIP profiles do not result in the strong enrichments and prominent peaks that are usually observed from ChIP-seq profiles of transcription factors or well-defined chromatin marks. We would like to emphasize that this low enrichment is an inherent observation for proteins with broad binding profiles, which — as in the case of DNMTs — have to interact with the entire genome in order to methylate every CpG. As we already mention in the manuscript (page 6), this results in a global binding of DNMTs to the entire methylated genome following CpG density, but excluding active promoters, enhancers and CpG islands. The preferential recruitment of DNMT3A1 to H3K27me3-marked CpG island shores, or DNMT3B to H3K36me3-positive gene bodies (as previously reported) is an additional binding preference on top of the global binding to the genome. Therefore the enrichment at these additional sites in comparison to the remainder of the genome is expected to be moderate.

We have now added additional information to Figure 1D and S3E to better highlight the differences between DNMT3A1 and DNMT3A2. We included now both tracks obtained from two different clones expressing DNMT3A1 and also a delta track that better visualizes the difference between DNMT3A1 and DNMT3A2. We hope that these additions are more convincing in regards to differential binding between the individual proteins — which was our initial intention to highlight in these examples.

- Is the apparent depletion of mC flanking DNMT3A1 binding sites an artefact of combining binding sites at the 5' and 3' of CGIs, i.e. If CGI 5' binding sites were separated from 3' binding sites, would the depletion look symmetrical or directional?

Response 10: The depletion of DNA methylation flanking the binding sites of DNMT3A1 (Figure 2D) is indeed occurring from CpG islands located either upstream or downstream. We have now reformulated the accompanying text in the manuscript to be more clearer: "... suggesting that genomic regions with reduced DNA methylation occur upstream or downstream of DNMT3A1-binding sites". As suggested by the referee, we have now also reanalyzed the DNA methylation around DNMT3A1 binding sites based on the site preference and orientation of the nearest CpG island. These new plots indeed reveal that the CpG island position upstream or downstream of the DNMT3A1 binding site is responsible for the depletion in DNA methylation (new Figure S4D), however there is no apparent difference in depletion of DNA methylation between DNMT3A1 binding sites flanking 5' or 3' ends of oriented CpG islands.

- Fig 2e, have CpG island promoters / TSSs been oriented so that all are represented at the 5' of genic regions, i.e. in the same direction as the associated gene? The symmetry of the H3K36me3 signal around the promoter CpG islands would suggest that CpG islands on the Crick strand have not been reversed, compared to the H3K36me3 asymmetry observed in Fig 4a. If not already done, it would be useful to orient all in the same direction relative to the gene coding direction so that any asymmetric features can be discerned.

Response 11: The reviewer is correct in pointing out that the CpG islands in Figure 2E have not been oriented according to the overlapping promoters. These CpG island were retrieved using the CpG cluster algorithm which just utilized DNA sequence information to call CpG islands (Hackenberg et al. 2006, PMID 17038168) and did not have any information regarding the orientation of the underlying gene. We have now retrieved this information from the underlying gene promoters and have re-calculated the heat maps as requested (New Figure 2E)

- It can be challenging to interpret how broadly a pattern is observed when only metaplots are used that aggregate signal over all members of a set of genomic regions. For example, from Fig 5a it is not clear how broadly the reintroduction of DNMT3A1 into the TKO cells restores DNA methylation at DNMT3A1 sites or bivalent CpG islands. Are all regions restored to the same degree, or is there heterogeneity, and if so, is it related to any other features? It would be very helpful to also include heatmap representations of the signal at these regions, as done in Fig 4a, even if only included in the supplement due to space restrictions, so that this could be explored further. This would also benefit other analyses e.g. those presented in Fig 4e, 5b, 5d.

Response 12: We have now included the requested heat maps for DNA methylation to Figures 5A (WT) and to Figure S10C to represent restoration of DNA methylation around CpG islands in the TKOs expressing DNMT3A1 and DNMT3A2. However, for the latter, we would like to emphasize that reintroduction of DNMTs to TKO cells results in sparse DNA methylation signals that are strongly diluted throughout the genome, and on average these CpGs have only 7-9 % methylation - which strongly affects their representation by heatmaps. This was also the main reason we have not included these heatmaps in the initial submission. In order to visualize these datasets we had to strongly reduce the threshold for the heat map representation in order to make the methylation at individual CpGs visible (we mention this in the legend of Figure S10C). From these heat maps we observe that *de novo* methylation in the DNMT3A1-reconstituted TKO cell line is more pronounced around CpG islands enriched for DNMT3A1.

We have also tested if other genomic and chromatin features could contribute to isoform-specific *de novo* DNA methylation activities and 5hmC deposition in the TKO cells expressing DNMT3A1 or DNMT3A2 - which we have now conducted in a genome-wide manner and also around enhancers (new Figure S11C and D). However we do not observe any other significant contribution to isoform specificities from the histone

marks analyzed, rather than that of H3K27me3 in relation of DNMT3A1 (Figure S11C), and also no isoform-specific roles of DNMT3A at enhancers (Figure S11D and E) - suggesting that the differential binding and activity is only restricted to H3K27me3 CpG island promoters.

- ChIP-seq of H3K27me3 should be performed in TKO, TKO + DNMT3A1, and TKO + DNMT3A2 cells in order to determine in this system whether loss of DNA methylation results in spreading of H3K27me3 and central depletion, and whether this spreading would be limited or prevented upon restoration of DNMT3A1 activity, but not DNMT3A2.

Response 13: These experiments have been performed in a previous study from the Fang Lab (King et al 2016, PMID 27681438) in which the authors report spreading of H3K27me3 in *Dnmt*-TKO and *Dnmt3a*/*Dnmt3b*-double-KO (DKO) cells, and partial resetting of H3K27me3 following re-introduction of individual *de novo* methyltransferases — especially by the DNMT3A1 isoform. In the initial submission we have already referred to this study and also performed re-analysis of their datasets. We have now extended this analysis based on the published datasets.

We show that the CpG island promoters enriched for DNMT3A1 binding show the strongest reduction in H3K27me3 in both *Dnmt*-TKO and *Dnmt*-DKO cells (Sup. Figure S12A). Concomitant with the decreased H3K27me3, we observe increased H3K27ac, chromatin accessibility and transcriptional activity at these CpG island promoters in *Dnmt*-TKO cells (new Figures 6E, S12D to G) — in line with previous studies suggesting that Polycomb-regulated CpG island promoters are frequently activated in absence of DNA methylation (Fouse et al 2008 PMID 18371437; King et al 2016 PMID 27681438).

By re-analysing the datasets from the re-constituted *Dnmt*-DKO and *Dnmt*-TKO cells provided by King et al., 2016, we observe that the H3K27me3 signal at DNMT3A1-bound CGI promoters is indeed restored upon reintroduction of DNMT3A1, and less effectively upon reintroduction of DNMT3A2 or DNMT3B (Figure S12 B and C). However, as also noted in our response to Reviewer 1 (Res.#5) and in line with the observations of King et al., strongest restoration is observed only in the DKO cells, but not in the TKO cells (Figure S12B and C). This is apparently a consequence of the missing DNA methylation maintenance activity in the TKO cells, where the established *de novo* methylation is diluted every cell division. This results in insufficient re-methylation and therefore incomplete resetting of H3K27me3 upon expression of DNMT3A1 in TKO cells.

- The increased 5-hmC production around bivalent CpG islands is not clear from Fig 5c. This should be clarified in the figure, or genomic regions that more clearly exemplify these patterns should be presented.

Response 14: We have highlighted the regions around CpG islands with increased 5-hmC production in Figure 6D (previously Fig 5c). As noted above, reintroduction of DNMT3A1 or DNMT3A2 to TKO cells results in reduced cytosine methylation that is furthermore depleted with each cell division in absence of DNMT1. Therefore the substrate for TET-mediated oxidation is limited, resulting in moderate 5-hmC accumulation that is not fully comparable with the levels measured in WT cells. Nevertheless, we provide now replicate measurements for 5hmC in TKO cells expressing DNMT3A1 or DNMT3A2 resulting in identical 5-hmC deposition, supporting the DNMT3A1-specific increase of 5-hmC at H3K27me3 positive sites (Figure 6C and S10F).

- More details about library characteristics and quality should be provided in the supplement, such as number of reads, mapping rate, coverage (where appropriate), and bisulfite conversion quality/frequency.

Response 15: We have now added a supplemental table containing this requested information (Table S1). This lists the number of sequenced reads and the number of reads uniquely mapped to the genome for ChIP-seq and 5hMeDIP-seq. For WGBS and RRBS, we have added the number of sequencing reads and the number CpGs covered more than 10 times. Furthermore, we have calculated the conversion rates based on spiked-in genomic DNA from unmethylated Lambda and *in vitro*-methylated T7 phage DNA (See also Figures S9D and S10A).

- Fig 1c, should show DNMT3a2 binding over genes too

Response 16: We have now included the density profile for DNMT3A2 to Figure 1C.

Referee #3:

DNA modification at cytosine residues is a prevalent feature of vertebrate genomes and its presence at regulatory elements can constitute an additional layer, in concert with gene regulatory networks (GRNs), for control of gene expression. This represents an entry point for DNA modification in processes such as X inactivation, repeat sequence inactivation, imprinting, and differentiation. We also know that DNA methylation landscapes impacts on other chromatin modifying activities and can thus influence gene expression states indirectly.

The DNA methylation machinery in mouse embryonic stem cells (mESCs) is intriguing as it has multiple components; multiple de novo methyltransferase isoforms, co-factors (Dnmt3l, UHRF1, Sirt1) and methylcytosine oxidases (Tet enzymes) that maintain a dynamic pattern of methylation. Yet fully hypomethylated mESCs are viable. Here Manzo and colleagues report that a specific de novo isoform, DNMT3A1 preferentially localises to the methylated shores of bivalent CpG islands (CGI), whereas its shorter isoform DNMT3A2 is globally distributed throughout the genome. They suggest that DNMT3A1 is required to protect CpG island shores from hypomethylation by counteracting TET-mediated oxidation of methylated cytosine. For the latter point, it could be equally argued that TET-mediated oxidation of methylated cytosine protects CpG island shores from hypermethylation. This illustrates that it is difficult to draw substantive conclusions from the experiments presented.

I think the data presented is of a high quality and expertly analysed to an excellent standard. The authors should be congratulated on this aspect, but I have some comments that need to be addressed.

Response 17: First we would like to thank the reviewer for acknowledging and highlighting the quality of our work. Regarding our conclusions, we fully agree with the reviewer that TET-mediated oxidation of methylated cytosines protects CpG island shores from hypermethylation. We think that both DNMT3A1 and TET proteins are required to fine-tune the methylation state of the CpG island flanks, and as stated — it is hard to discern who protects from what during the turnover of methylcytosine. We have now changed the respective text passages in the manuscript to be more clearer on this aspect.

(A) Experimental:

1. The experiments essentially depend on an over expression system of tagged proteins based on RCME insertion of constructs driven from the CAGGs promoter in mESCs and derived cells. What is the evidence that this system mimics endogenous expression of the particular isoforms and the resulting 5mC patterns? A long and considered answer is expected for this question.

Response 18: The reviewer raises the valid concern that the additional expression of DNMT3A transgenes from the CAG promoter would influence the results obtained from the genome-wide binding analysis. As also mentioned in response #3 to Reviewer 1, the reason for using the CAG promoter is to prevent loss of transcription of the biotin-tagged proteins from the RMCE site during long-term cultivation or differentiation of ES cells. While this does not mimic cell-specific expression levels of the endogenous gene, it is important to emphasize here that we express the transgene from a single locus on one allele only. In contrast to strong transgene “overexpression” levels observed from transient or random integration experiments which usually result in multiple copies of the same transgene expressed per cell, our additional expression system from a single locus, even with CAG promoters, leads to modest increase in protein levels (see below). Furthermore, in order to be able to accurately compare and understand the binding preference of individual DNMT3A isoforms — which was the major aim of this study — we require comparable expression of both isoforms. In addition, we show that expressing the biotin-tagged proteins at lower levels does not change their binding preferences (see below).

Based on Immunoblot analysis (Figure S2C), the levels of the tagged isoforms in comparison to the endogenous proteins are indeed higher, by approximately 2 to 3-fold. However, based on this Immunoblot analysis we cannot make reliable statements about the protein levels. As part of our efforts to quantify this heterologous expression, and to test if expression level influences binding, we have now performed parallel reaction monitoring (PRM) mass spectrometry to measure DNMT3A protein levels in a targeted manner directly from nuclear extracts. This allows us to accurately identify and quantify specific DNMT3A peptides from trypsin-digested nuclear extracts, and compare their abundance between wild type ES cells and cells expressing either biotin-tagged DNMT3A1 or DNMT3A2. For this we have used 4 different peptides that are shared between the DNMT3A isoforms, and measured them in four independent replicates each. These new results which we present now (in Figure S2D and S2E) indicate that the total abundance of DNMT3A

(tagged and endogenous DNMT3A together) in these cell lines is increased by 2.6-fold in the DNMT3A1 clone_1 and by 2.5-fold in the DNMT3A2 cell line, when compared to wild type cells (1.4 and 1.29 in log₂-FC). We now mention this increase in protein levels in the manuscript (page 5). While this is indeed above the endogenous expression — we argue that the increase in protein levels is rather moderate and should not influence the results obtained in this study.

In order to address if this 2.5x increase in expression would influence the genomic binding of the tagged DNMT3A proteins, we have included a bioChIP-seq dataset that was obtained from DNMT3A2 transgene expressed under the control of a weaker promoter (CMV). These were performed in a previous study where we have addressed similar concerns (Baubec et al., 2015). While the expression of the CMV DNMT3A2 protein is much lower compared to its CAG-expressed counterpart, its genome-wide binding remains similar to the binding observed from the DNMT3A2 proteins under CAG promoter control (please see Rebuttal Figure 1 in response to Reviewer 1, and Figure S3B in the manuscript).

Furthermore, we have included two individual clones of biotin-tagged DNMT3A1. As can be noted from the Immunoblot in Figure S2C where we detect the biotin-tagged proteins using Streptavidin-HRP, the two individually-derived clones of DNMT3A1 show different expression levels, whereas clone 2 is much lower expressed than clone_1. The higher-expressing clone_1 was also used to quantify DNMT3A levels in the PRM experiments above. Despite this difference in expression levels, we do not observe any differences in genomic localization between these two cell lines - which is observed from genome browser tracks where we show both clones (Figure 1E and S3E) or genome-wide correlation analysis (Figures 1F and S3A,B and F).

Taken together, these results indicate that the CAG promoter expression from single-locus integration in ES cells results in moderate expression levels, and furthermore, our results obtained with transgenes expressed at lower levels indicate that this elevated expression of biotin-tagged DNMT3A isoforms does not influence their genomic binding. We hope these experiments and analysis steps address the concerns of the reviewer sufficiently.

2. Does over-expression of the particular isoform under test unbalance the system? For example is the stoichiometry between co-factors (e.g. Dnmt3l) and other partners (other Dnmt3s) when massive amounts of the exogenous protein are present, may this affect the DNA methylation pattern outcomes?

Response 19. The reviewer raises the concern that the additional expression of DNMT3A proteins could unbalance the stoichiometries between DNMT proteins. In line with the changes in DNMT3A protein levels measured by PRM, we were able to quantify additional proteins involved in regulation of DNA methylation, H3K27me₃ and additional proteins that served as control (new Figure S2D). As already discussed in Response 18, the additional expression of DNMT3A isoforms from the RMCE site leads to a mild, but not “massive” increase in protein abundance. In addition we can also observe a minor increase in DNMT3L peptides, whereas peptide abundance changed from 28.4 in wild type cells to 29.4 and 29.1 in cells expressing DNMT3A1 and DNMT3A2, respectively. This suggest that additional expression of DNMT3A leads to stabilization of DNMT3L, probably through protein-protein interactions, which we observe in the new co-immunoprecipitation experiments in Figure 4A and B. This mild increase suggests that the system stays in balance upon expression of additional DNMT3A copies. Other proteins that could be detected by PRM including: DNMT1, UHRF1, HELLS, EED, EZH2, RING1B and control proteins were not affected by the additional expression of DNMT3A.

We also observe that when we removed DNMT3A via CRISPR, expression levels of DNMT3B and DNMT3L detected by immunoblot remain unchanged in ES and neuronal progenitors (New Figure S9A). Taken together these experiments argue against a strong deregulation of the DNA methylation machinery upon heterologous expression and deletion of DNMT3A isoforms.

3. Does co-expression of isoforms (A1, A2, 3B or 3L) essentially give the same result as single gene over-expression?

Response 20. The reviewer raises an interesting question in regards to how stoichiometries of DNMT3 proteins could affect observed binding behavior of biotin tagged DNMT3A isoforms. Based on previous bio-ChIP experiments performed for DNMT3A and DNMT3B proteins in wild type cells and in *Dnmt*-TKO cells we do not see stark differences in the genomic binding patterns of the tagged DNMTs (see Baubec et al 2015,

Supplemental Figure 3). Furthermore, during ES cell differentiation where expression levels of endogenous DNMTs are altered (i.e. DNMT3B and DNMT3L are strongly reduced, see new Figures S1C for mRNA levels and S9A for Immunoblots) we do not see gross changes in the protein stability or genome-wide binding of the tagged proteins. The only difference we observe is related to H3K27me3 dynamics in case of DNMT3A1. These results already suggest that presence or absence of other DNMTs do not strongly contribute to the observed genome-wide binding patterns, and we conclude from these results that additional co-expression of other DNMTs would not influence the observed genomic binding patterns in our cellular system.

However, in order to test how co-expression of two or more DNMT proteins would influence their genomic binding, as suggested, would require an elaborate system with controlled expression and individual tagging of multiple proteins. While we fully agree that this would be a reliable setup to test how inter-DNMT interactions and stoichiometries affect DNMT binding and function, we think that these experiments would be beyond the scope of the current manuscript and not feasible in the allocated timeframe. We plan to follow up on these questions in more detail using well controlled genetic and biochemical studies.

4. One possible way to address these questions (2 and 3) is to use CRISPr technologies to tag endogenous genes (either with the Bio or another tag) and validate if the ChIP patterns match the over-expression results.

Response 21. We thank the reviewer for this suggestion. We have indeed attempted to tag the individual endogenous DNMT3A isoforms several times in the course of this study using various CRISPR and TALEN-based technologies — however without success. This has mostly to do with the *Dnmt3a* gene structure and where the ATGs of the individual isoforms are located. DNMT3A2 for example, cannot be tagged without disrupting the ORF of DNMT3A1 (the DNMT3A2 ATG lies within the coding exon 6 of DNMT3A1). On the other end, the C-terminal ends of both isoforms are identical which would not allow us to distinguish between the isoforms. We have also performed several attempts to add the biotin tag to the ATG of DNMT3A1, however due to the high GC% at this region (70-90%) the homologous recombination of the biotin tag donor failed so far.

We hope that in the near future we will be able to solve this issue since we agree that mapping the endogenous proteins would be the ultimate validation for our results. However, as already mentioned in Response 18, the expression of the biotin-tagged DNMTs in our system is based on a single-copy integration in the genome, and the resulting expression levels are not comparable with overexpression observed from transient expression or multi-copy integration experiments. Furthermore, by comparing the two clones that display different expression levels of DNMT3A1 (clone 1 = high, clone 2 = low, see Figures 1E, 1F, S2C, S3A and S3B), and also by reducing the expression level of DNMT3A2 through the use of a CMV promoter, we have already tested the influence of expression levels on genomic binding. These experiments do not indicate any influence of expression levels on the genome-wide targeting preference of the DNMT3 proteins. We hope that these arguments convince the reviewer about the validity of our results.

(B) The manuscript and experimental:

The authors state that an illustration on how important DNMT3A methylation is, is the phenotypes of mice with mutant DNA methyltransferases. However, this does not square with viable hypomethylated mice resulting from mutations in co-factors such as HELLS (very hypomethylated) or UHRF1 (10% reduction in global methylation). The authors should adjust the introduction accordingly as the phenotypes of hypomethylated mice depend on their derivation route. Given their conclusions, it should be noted that DNMT3A KO mice are sub-viable; they get through early development fine. DNMT3A^{-/-} mice develop to term and appear to be normal at birth. However, most homozygous mutant mice become runted and die at about 4 weeks of age, long after the potential events described in this report.

Response 22: We thank the reviewer for pointing this out. We have indeed missed to mention differential phenotypes in regards to the individual *Dnmt*-KOs in our introduction, which we do now (page 3).

The role of the new isoform, DNMT3C, should be mentioned in the introduction, what are their expectations of how this DNMT3 isoform will perform in their system?

Response 23: We have now mentioned DNMT3C and its role in regulating DNA methylation during rodent germline development in the introduction (page 4). So far, we can only speculate about how the DNMT3C

paralog binds to the genome. However, since the focus of this study is DNMT3A, we would rather omit these speculations since these would unnecessarily extend the discussion.

Primary citations should be used for the statement that 'DNA methylation is highly dynamic and undergoes constant turnover at regulatory sites'. I suggest that the following are very appropriate: PMID 26928226 and 27346350.

Response 24: We thank the reviewer for pointing this out and suggesting references. We now cite Stroud et al., 2011 PMID 21689397, and Feldmann et al, 2013 PMID 24367273, which we consider to first suggest that 5hmC presence at promoters, enhancers and transcribed gene bodies reflects a regulatory turnover of DNA methylation. We hope the reviewer agrees that these citations are appropriate.

The statement 'H3K9me3 plays a role in maintenance of DNA methylation via the accessory protein UHRF1 (Rothbart et al,2012; Meyenn et al, 2016)' is erroneous because Meyenn links H3K9me2 with maintenance methylation and this can be challenged by PMID 27554592, which concludes that 'while our study (with a Uhrf1 knockin (KI) mouse model) supports a role for H3K9 methylation in promoting DNA methylation, it demonstrates for the first time that DNA maintenance methylation in mammals is largely independent of H3K9 methylation.'

Response 25: We agree with the referee that the connection between H3K9me3 and DNA methylation has not been fully resolved in mammals. For the sake of readability we have removed this passage and the reference to H3K9me3.

The last paragraph in the introduction appears to be a reproduction of the abstract; can they be a bit more informative?

Response 26: We have changed both the abstract and have added additional information to the last paragraph of the introduction.

Regarding the CAGE data, can the authors expand on this to include DNMT3C, DNMT1S, DNMT1O and DNMT3L. With respect to their proposal regarding 5hmC, a TET isoform CAGE survey would also be informative. Their survey should include as many stages (including adult) as possible and tissue types.

Response 27: As requested, we have now added more information from the FANTOM consortium regarding the expression of other DNA methylation -related proteins (DNMT1, DNMT3B, DNMT3L and TET1-3) and their isoform-specific expression in various stages and tissues (See new Figure S1E and the provided Source data that contains the RPM values for each promoter in all samples). It is however worth mentioning that not all isoforms can be analyzed in this manner: some are based on alternative splicing and utilize the same promoter (e.g. DNMT3B isoforms), and for some proteins that utilize alternative promoters to generate isoforms not all promoters could be detected by CAGE-seq (e.g. TET1 or DNMT1).

The IHC in Figure S1D is of poor resolution and lacks sizing bars, a better picture would be appreciated.

Response 28: We have now replaced the initial IHC figures by new data showing biotin-tagged DNMT3A1 localisation in ES and neuronal progenitor cells. We have also included the sizing bars as requested (see new Figure S2D)

While the 5hmC argument is interesting, it is not compelling as it appears correlative and not causative. What happens if Tets are inactivated in the DNMT3A1 over-expressing mESCs, do you get methylation creep at the coastal regions of the CGIs?

Response 29: We would like to refer to a recent publication from Kong et al., 2016 (PMID 27288448) to answer this question. In this study, the authors have found that deletion of TET1/2 in ESC results in decreased 5hmC and increased 5mC at H3K27me3 positive CpG islands. This indicates that TETs are required to protect from DNA methylation spreading into the CpG island. However, this spreading is not complete due to the presence of H3K4me3 in the center of the CGI and repulsion of DNMT3A proteins. Similar results were obtained in hematopoietic stem cells by Zhang et al 2016 (PMID 27428748), identifying that *de novo* methylation at CpG islands frequently occurs in absence of TET2.

Similar to what we have discussed in Response 17, we think that both enzymes (TETs and DNMT3A) are necessary to balance the level of DNA methylation at the CpG island shores — one ensuring that these sites stay methylated, the other one that methylation does not spread in. We have modified our arguments throughout the manuscript to be more clearer.

Figure 4a requires a 5mC heat-map profile in addition to 5hmC.

Response 30: We have added the requested mCG heat map to figure 4a

During NPC derivation, how does expression of the endogenous DNA methylation machinery change (DNMT-1,-3A,-3B,-3C and 3L; Tet1, 2 and 3), how will this impact on the outcome of over-expressing DNMT3 isoforms in this system compared to mESCs?

Response 31: Similar to our response to Reviewer 1 (see response #2), we provide now various quantitative measurements for endogenous DNMT proteins in ES and NPC. Regarding DNMT3A1 and DNMT3A2, we indicate now by qRT-PCR, Immunoblot and by RNA-seq profiles that both isoforms show an increased expression in neuronal progenitors, when compared to ES cells (new Figures S1A and B). Furthermore, we have included Immunoblot detection of DNMT3B and DNMT3L showing decreased expression for these proteins in NPC (Figure S9A). In addition we provide now quantification of gene expression changes during differentiation for all relevant members of the DNA methylation machinery based on microarray data (new Figures S1C) - as requested by the reviewer. DNMT3C could not be included since there are no probes on this array.

We have also investigated a possible influence of these changes on the stability of our biotinylated proteins. Based on immunoblot detection we do not observe drastic changes in the levels of the biotin-tagged DNMT3A proteins during differentiation (new Figure S2C). Furthermore, genome-wide binding remains largely similar between ES and NP cells, with the exception of DNMT3A1 relocation to H3K27me3 sites.

Taken together we conclude from these results that the protein dynamics observed for many endogenous DNMT proteins during differentiation do not influence the stability or binding profiles of the biotinylated proteins.

The data in figure 5c-d, must be interrogated in the context of the recent publication by Xiong et al (2016: PMID-27840027), who suggest that there is collaborative interaction between SALL4A and TET proteins in regulating stepwise oxidation of 5mC at enhancers. What is the profile of enhancer 5hmC in the TKO-DNMT3A1 and TKO-DNMT3A2 rescue cell lines? Is this restored without requirement for DNMT1?

Response 32: We thank the reviewer for this suggestion. We have now focused our analysis on enhancers and the dynamics of DNA methylation and 5hmC at these sites in TKO cells expressing DNMT3A1 or DNMT3A2. While we observe that 5hmC levels are partially regained by expression of DNMT3A1 or DNMT3A2 in TKO cells, these do not reach the initial levels observed in WT cells and do not show any differences between the introduced DNMT3A isoforms (new Figure S11E). In addition, *de novo* methylation in TKO cells expressing DNMT3A1 or DNMT3A2 is partially regained around these sites, with the expected depletion of mCG in the center of the enhancer element (new Figure S11D). In addition, we do not observe isoform-specific *de novo* methylation differences at enhancers - besides the difference in global re-methylation observed outside of the analyzed elements (new Figure S11D). This suggests that, while DNMT3A partially contributes to *de novo* methylation around enhancers and therefore increases 5hmC production at these sites, there is no isoform-specific preference detected.

We furthermore, now also provide new analysis about additional 5mC oxidation steps and their accumulation in absence of TDG at sites preferentially bound by DNMT3A1. These new results indicate that in absence of TDG, 5-fC and 5-caC levels are substantially increased around DNMT3A1 binding sites, further emphasizing the role of DNMT3A1 in catalyzing the TET-mediated oxidation (new Figures S11A and B).

I was bit surprised that Armand et al (2012: PMID- 22761581) was not discussed, who identified a substantial incomplete regional methylation maintenance and importantly that non-CpG cytosine methylation is confined to ESCs and exclusively catalysed by DNMT3A and DNMT3B, is there any difference in non-CpG

methylation in their various rescued cell lines? Is non-CpG methylation accentuated the transgenic mESCs and NPCs?

Response 33: We have now included non-CpG methylation analysis from the WGBS data obtained in the *Dnmt*-TKO cell lines re-expressing DNMT3 proteins (New Figure S10B). We represent these results as bar-plots that indicate the methylation frequencies at CG, and various CHG and CHH sites of chromosome 19, whereas H stands for A,C,T. We confirm that both, DNMT3A and DNMT3B contribute to non-CpG methylation (in accordance with Arand et al, and also earlier publications by Bird and colleagues which we cite now). However, besides differences in global methylation levels, we do not observe stark differences in non-CpG methylation preference between the rescued cell lines. We thank the reviewer for suggesting this analysis.

Since we have not performed WGBS in the transgenic wild type ES and NPC cell lines, we cannot make any statements about changes in non-CpG methylation in these cells at this time. For this we would require very deep coverage WGBS maps for all cell lines and stages used in this study in order to accurately detect small changes in non-CpG methylation. While we appreciate this suggestion and agree that it would be interesting to see differences in non-CpG DNA methylation upon expression of additional DNMT copies in these cells, we believe that this would be a very cost-demanding endeavor that is beyond the current scope of this study. We hope the reviewer agrees to that.

Thank you for submitting a revised version of your manuscript. It has now been seen by two of the original referees whose comments are shown below.

As you will see they both find that all criticisms have been sufficiently addressed and they recommend the manuscript for publication. However, before we can go on to officially accept the manuscript there are a few minor editorial issues concerning text and figures that I need you to address in a final revised manuscript:

-> I noticed that you currently have all supplemental information placed in the Appendix file. You're welcome to take advantage of our Expanded View format and 'promote' up to five of these to EV figures that will be typeset and displayed in-line with the main manuscript in the html file (<http://emboj.embopress.org/authorguide#expandedview>). Please make sure to update nomenclature and call-outs accordingly.

-> In addition, please format the Appendix file according to our author guidelines (including calling it 'Appendix' instead of 'Supplementary data'). Ideally, also add 'Appendix' in front of all items in the TOC (Appendix Supplementary Methods, Appendix References) and add 'Appendix' to the corresponding titles.

-> We also noticed that all appendix figures are very small and the text in them is hard to read. Since there is no page limit, you are welcome to make the figures/text bigger.

-> The files currently labeled Source Data seem to be data set files rather than classical source data. I'd encourage you to relabel them as Datasets, include a legend for each of them in a separate tab, and to change the call-outs accordingly. In addition, I'm not sure how the proteomics data labeled 'Source_data_FigS2C' relates to the Western blot shown in that figure panel. Feel free to contact me with any questions about this.

-> In addition, please amend the following missing call-outs:

- All callouts for Appendix figs are missing an S, example: 'Appendix Figure 1A' should be called out as 'Appendix Figure S1A'.
- Callouts are missing for the following panels: Appendix figs S2E, S3C &D, S5G, S7F.
- Appendix figs S8 and S9 have been called out, but none of the individual panels in S8 (A-D) and not S9D.
- Callout missing for Appendix Table S1.

Thank you again for giving us the chance to consider your manuscript for The EMBO Journal, I look forward to receiving your final revision.

REFeree REPORTS

Referee #2:

The authors have added substantial new experiments and analyses in their revised manuscript, which significantly improves the study and suitably addresses all of my initial comments. This is a high quality and interesting study that provides new insights into the roles of DNMT isoforms and their function at bivalent CpG islands.

Referee #3:

This is a revision of a manuscript (EMBOJ-2017-97038) submitted to EMBO, in which a number of queries were raised by myself and other referee's.

Manzo and colleagues report that a specific de novo isoform, Dnmt3A1 preferentially localises to the methylated shores of bivalent CpG islands (CGI), whereas its shorter isoform Dnmt3A2 is globally distributed throughout the genome. They suggest that Dnmt3A1 is required to protect CpG island shores from hypomethylation by counteracting TET-mediated oxidation of methylated cytosine or alternatively that TET-mediated oxidation of methylated cytosine protects CpG island shores from hypermethylation.

I think the authors have made huge efforts to satisfactorily address all the extensive critiques raised by the referees and that their conclusions are justified. I have no further comments except to say 'well done' to the authors.

2nd Revision - authors' response

01 October 2017

We thank you for the opportunity to submit our revised manuscript where we have addressed all editorial remarks.

We have now included three Extended View Figures to the manuscript and have reformatted the Appendix file according to your suggestions. Furthermore we have resized the figure panels and text in the Appendix Figures.

The previous files submitted as Source Data are now labeled as Datasets_EV1 to 3 and submitted together with a README.txt file containing relevant information about the contents. Datasets are now called out in the manuscript.

All missing callouts to Appendix figures have been included in the text and in the required format.

We hope these changes address the requests sufficiently.

3rd Editorial Decision

03 October 2017

Thank you for submitting the final version of your manuscript, I am pleased to inform you that the study has now been officially accepted for publication in the EMBO Journal.

EMBO PRESS

YOU MUST COMPLETE ALL CELLS WITH A PINK BACKGROUND ↓
PLEASE NOTE THAT THIS CHECKLIST WILL BE PUBLISHED ALONGSIDE YOUR PAPER

Corresponding Author Name: Tuncay Baubec
Journal Submitted to: EMBO Journal
Manuscript Number: EMBOJ-2017-97038